# ARF suppression by MYC but not MYCN confers increased malignancy of aggressive pediatric brain tumors

Oliver J. Mainwaring ®[1,4], Holger Weishaupt[1,4], Miao Zhao ®[1], Gabriela Rosén[1], Anna Borgenvik[1], Laura Breinschmid[1], Annemieke D. Verbaan[1], Stacey Richardson[2], Dean Thompson[2], Steven C. Clifford[2], Rebecca M. Hill[2], Karl Annusver ®[3], Anders Sundström[1], Karl O. Holmberg[1], Maria Kasper ®[3], Sonja Hutter[1] & Fredrik J. Swartling ®[1] ✉

Medulloblastoma, the most common malignant pediatric brain tumor, often harbors *MYC* amplifications. Compared to high-grade gliomas, *MYC*-amplified medulloblastomas often show increased photoreceptor activity and arise in the presence of a functional ARF/p53 suppressor pathway. Here, we generate an immunocompetent transgenic mouse model with regulatable *MYC* that develop clonal tumors that molecularly resemble photoreceptor-positive Group 3 medulloblastoma. Compared to *MYCN*-expressing brain tumors driven from the same promoter, pronounced *ARF* silencing is present in our *MYC*-expressing model and in human medulloblastoma. While partial *Arf* suppression causes increased malignancy in *MYCN*-expressing tumors, complete *Arf* depletion promotes photoreceptor-negative high-grade glioma formation. Computational models and clinical data further identify drugs targeting MYC-driven tumors with a suppressed but functional ARF pathway. We show that the HSP90 inhibitor, Onalespib, significantly targets MYC-driven but not MYCN-driven tumors in an ARF-dependent manner. The treatment increases cell death in synergy with cisplatin and demonstrates potential for targeting MYC-driven medulloblastoma.

Brain tumors are the leading cause of cancer-related deaths in children, with medulloblastoma (MB), the most common malignant pediatric brain tumor, comprising almost two thirds of all embryonal pediatric brain tumor cases[1]. MB is stratified into four molecular subgroups: WNT, SHH, Group 3, and Group 4, each with their own intrinsic genetic and molecular landscapes[2]. *MYC* family genes are often overexpressed in MB, and *MYC* or *MYCN* amplifications are correlated with poorer prognoses[3]. MYC and MYCN show high structural homology that translates into a highly conserved function where the genes are able to substitute each other during development[4]. *MYC* family gene amplifications are often mutually exclusive where *MYC* amplifications are more specifically found in Group 3 tumors while *MYCN* amplifications can be found in SHH, Group 3, or Group 4 tumors[2]. Most Group 3 tumors further show high activity of photoreceptor-specific transcription factors that further defines important molecular subsets of this subgroup[5–7].

Patients are stratified into low-, standard-, and high-risk groups contingent on presence of metastatic seeding, genetic risk markers,

[1]Department of Immunology, Genetics and Pathology, Science for Life Laboratory, Rudbeck Laboratory, Uppsala University, Uppsala, Sweden. [2]Wolfson Childhood Cancer Research Centre, Translational and Clinical Research Institute, Newcastle University Centre for Cancer, Newcastle upon Tyne NE1 7RU, UK. [3]Department of Cell and Molecular Biology, Karolinska Institutet, 171 77 Stockholm, Sweden. [4]These authors contributed equally: Oliver J. Mainwaring, Holger Weishaupt. ✉e-mail: fredrik.swartling@igp.uu.se

and defined molecular subgroups; and treatment is tailored in respect to their risk stratification[8]. Overall survival of MB is ~70–80% but patients are often left with serious late-sequelae, a result of the sub-optimal, aggressive treatments causing collateral effects[9]. Survival is also highly dependent upon the subgroup being treated where meta-static MYC-driven Group 3 tumors are considered very high risk with <50% survival. Despite best efforts to understand the individual sub-groups, MB treatment remains rather uniform across subgroups and a more tailored treatment regimen would be highly beneficial.

Several recently generated *MYC*-driven MB models use viral overexpression and orthotopic transplantation or in utero electro-poration as a means for overexpressing *MYC* in potential cerebellar cells of origin of MB[10–16]. Transgenic models have the advantage of carefully modeling the tumor initiation process in cells within their natural environment and in an intact immune system in the developing brain. We previously described a regulatable, transgenic Tet-OFF model of Group 3 MB in which aggressive tumors are generated when *MYCN* expression is directed from a Glutamate transporter promoter 1 (Glt1) – the GTML model[17].

The expression of *MYC* and *MYCN* genes is regulated by efficient feedback and failsafe mechanisms. In normal cells, rapid onset of elevated MYC protein levels leads to cell death and induces apoptosis. One such failsafe program and important mediator of MYC-induced apoptosis is the ARF/p53 pathway[18]. The *p53* tumor suppressor gene and/or the *CDKN2A* gene, which encodes both ARF and INK4A, are commonly lost in high-grade gliomas (HGGs)[19,20] but rarely inactivated in medulloblastoma, and then solely in the SHH subgroup[21]. MYC is known to induce ARF levels and *ARF* is often silenced or mutated in cancer and/or is negatively selected during MYC-driven transformation[22,23]. This is in line with the fact that ARF further inhibits MYC-driven transformation in a p53-independent manner[24]. Benanti et al. have previously reported that *MYC* overexpression drives down-regulation and methylation of *ARF* and this results in immortalization of human cells[25]. MYCN directly inhibits p53[26] but it is less clear if MYCN can modulate ARF activity during brain tumor development. As cancer development is considered a multi-step process, it is possible that the two MYC homologs use divergent paths to modify the ARF/p53 pathway in order to generate distinct tumor entities. This has recently been described in transplanted MB models where distinct differences in how MYC and MYCN bound to MIZ1 were revealed and correlated to subgroups[27]. Still, these differences have not been studied in transgenic models where tumors evolve during development in an immunocompetent host.

With a deficit of knowledge underpinning MYC and MYCN-driven MB subgroups, genetic models are tantamount in understanding the key drivers involved, their genetic background, and potential avenues for treatment in patients. Here we present a transgenic model of Group 3 MB driven by *MYC* overexpression and portray the different levels of genetic regulation MYC exerts on ARF as compared to MYCN to drive tumorigenesis. We suggest that this regulation depends on functional ARF pathway activity and further identified inhibition of the chaperon protein HSP90 as a promising approach to target MYC-driven brain cancer.

## Results

### MYC expression in GLT1 + cells generates malignant brain tumors in the hindbrain

In Group 3 tumors, *MYC* amplifications are at least 3–4 times more common than *MYCN* amplifications[28]. In order to study the relationship of MYC and MYCN mRNA expression in MB we analyzed their expression levels in a large combined batch-normalized set of micro-array data from human MB samples and normal cerebellum[29]. As expected, MYC and especially MYCN levels were active and elevated in the developing normal brain but decreased with age (Fig. 1a). MYC levels were elevated in 60% of Group 3 tumors while MYCN levels were

only elevated in 2% of samples in this subgroup. Instead, 51% of SHH tumors showed high levels of MYCN. Obviously, when comparing the levels of MYC and MYCN it is clear that MYC is likely a more important oncogenic driver of Group 3 MB.

To clarify the extent of MYC involvement in the pathogenesis of medulloblastoma, we obtained a transgenic mouse line in which mice overexpress *MYC* under the TRE-operator, driven by the Tet system[30]. We crossed this TRE-MYC line with a tTA-expressing strain under control of the Glt1 promoter[17]. The Glt1-tTA/TRE-MYC (GMYC) Tet-OFF model could be regulated by doxycycline and was kept as doubly hemizygous crosses in the FVBN strain. Further development of the GMYC model was achieved by crossing it to include the TRE-CRE-LC1 luciferase construct[31], allowing all GMYC tumor cells to be monitored under luminescent imaging (Fig. 1b). Resultant brain tumors developed sporadically with an average latency of 90–150 days and about 60% penetrance. Tumor presentation was defined by a logarithmic increase in bioluminescent signal intensity, weight loss observed and commonly with a visible protrusion on the head of the mouse.

The inclusion of the tTA-Tet-OFF system allowed us to investigate the effect of *MYC* suppression on tumor maintenance through the administration of doxycycline (dox) into the diet of these mice. Tumor-presenting mice were put onto a regimen of dox-supplemented food for 30 days ($n = 8$). Loss of MYC led to complete tumor clearance and cure of these mice (Fig. 1c), confirming that these tumors are oncogene addicted. No mice showed any signs of relapse. Mice which did not receive any dox treatment died within a few days of tumor presentation.

Tumors developed in the hindbrain region adjacent to or within the cerebellum (Fig. 1, d, e). GMYC tumors histologically resembled GTML tumors, often presenting with an increased nuclear to cyto-plasmic ratio and large cell/anaplastic (LC/A) features, where tumor cells invading into the molecular layer are pleiomorphic and can be distinguished from the smaller, differentiated granule neurons (Fig. 1f). MYC was highly expressed throughout virtually all tumor cells, con-firming functional TRE-MYC activity in GMYC tumors (Fig. 1g). By using the GMYC/TRE-CRE-LC1 mouse, tumor expansion and progression in the brain can be followed with luciferase prior to phenotypic pre-sentation (Fig. 1H) and during regression of the tumor upon dox treatment (Supplementary Fig. 2g). As 40–60% of Group 3 patients present with leptomeningeal dissemination at diagnosis[2], it was imperative to check if an analogous phenomenon occurs in the GMYC model. In the GMYC model, unlike human Group 3 tumors, leptome-ningeal metastasis was a rare event, observed in 5% of cases ana-lyzed (Fig. 1i).

### Tumors resemble distinct human MYC-driven Group 3 MB subtypes

High levels of Synaptophysin and TuJ1 confirmed that the tumors had a more neuronal character and the intense OTX2 and NPR3 staining further suggested tumors were similar to non-SHH MBs (Fig. 2a). Tumors stained mostly negative for glial histological markers com-monly used for gliomas, such as GFAP and Olig2. Still, there were sometimes rare GFAP and OLIG2 + cells interspersed in the tumors that might reflect on a cellular heterogeneity that is also similarly found in human tumors. Tumors showed widespread proliferative activity, as visualized with Ki67 (Fig. 2a, b), and it was common to see tumor cells seeding out from the circumscribed tumor mass. Similar to human tumors, pockets of apoptosis were present in many areas of the GMYC tumors, as observed with strong Cleaved Caspase-3 positivity. This Cleaved Caspase-3 activity was not widespread throughout the whole tumor mass, but instead localized to these isolated pockets. (Fig. 2a, b).

Conventional RNA sequencing was performed on GMYC ($n = 6$) tumors that was compared to RNA sequences similarly generated on GTML ($n = 8$) tumors[32]. When compared via cross-species analysis to other human childhood brain tumors,

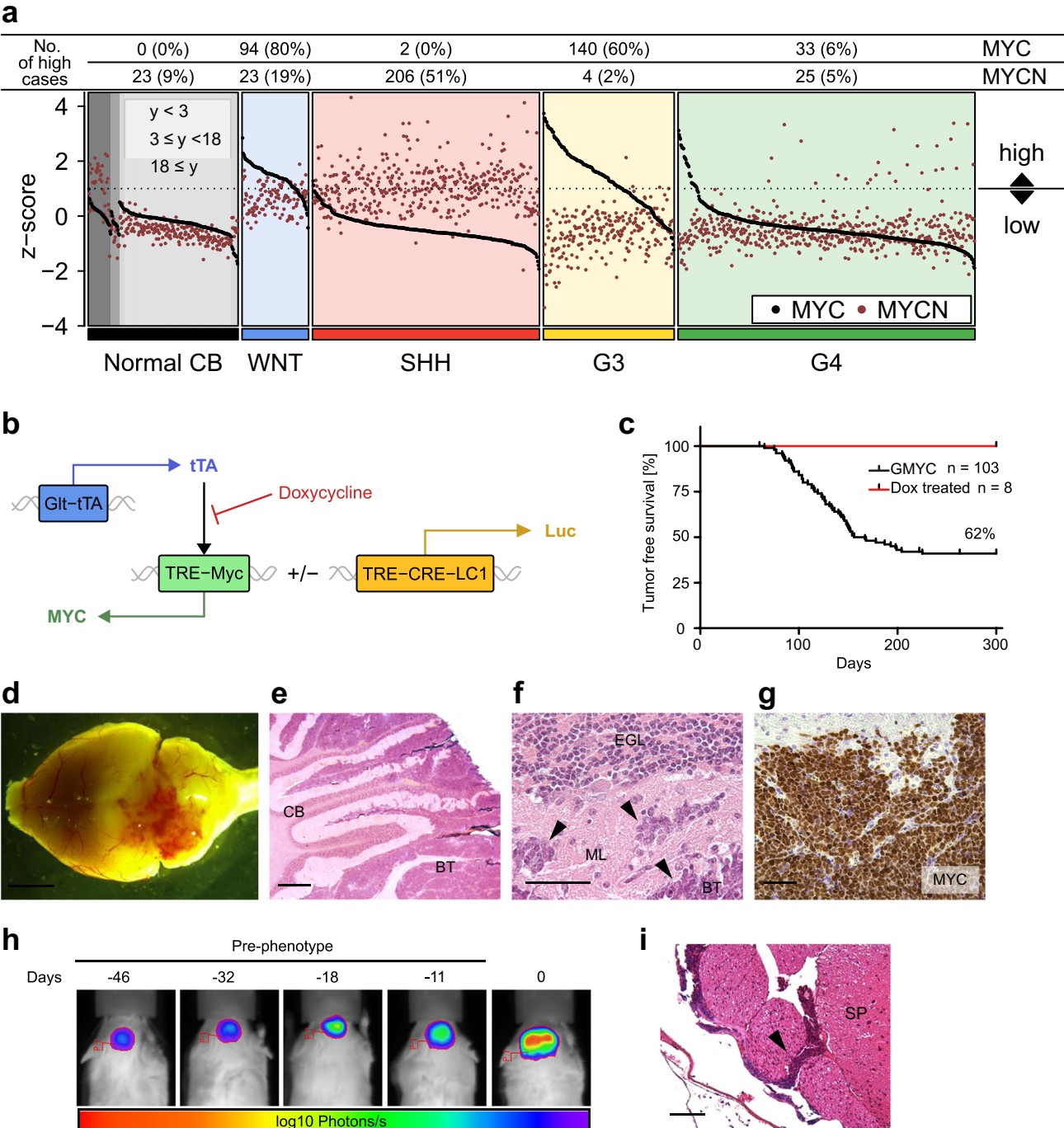

**Fig. 1 | MYC overexpression in GLT1 + cells generates malignant brain tumors in the hindbrain. a** Expression of *MYC* and *MYCN* across MB subgroups and normal cerebellum from infants (<3 years), children (3 ≤ years < 18) or adults (years ≥ 18). Expression is normalized across samples via z-score transformation. Samples are considered to exhibit high expression if *z* > 1. Data from (37). **b** Transgenic expression of *MYC* is driven through the TRE by tTA protein expressed by the Glt1 promoter. In presence of dox, tTA does not associate with TRE and transcription is ceased. Glt1-tTA-MYC mice could be bred to express the Luc1 gene under the same promoter. **c** Kaplan–Meier plot of GMYC survival (*n* = 103). GMYC mice treated with 30 days of dox (*n* = 8) at tumor presentation (red line) survived until experiment end at P300. Log-rank Mantel-Cox statistical test. **d** Gross overview of a representative GMYC mouse brain upon tumor formation. Scale bar represents 10 mm. **e** ×20 magnification of a GMYC cerebellum (CB) with brain tumor (BT). Infiltrative cells can be seen disrupting normal architecture of the cerebellum. Scale bar represents 500 μM. **f** ×40 magnification of MB cells invading the molecular layer (ML) of the cerebellum. Scale bar represents 100 μM. **g** Widespread MYC positivity throughout the brain tumor confirms overexpression of human MYC in the transgene. Scale bar represents 100 μM. **h** A GMYC/TreCRE-LC1 mouse followed (5 times) under IVIS starting 46 days before phenotypic tumor presentation (day 0). **i** H&E of a GMYC spinal cord displaying leptomeningeal metastasis (arrow). Scale bar represents 100 μM. H&E for **e**, **f** was performed at least once on all animals killed in **c** to confirm presence of tumor. In **g**, experimental data was verified in at least two independent experiments. G3: Group 3; G4: Group 4.

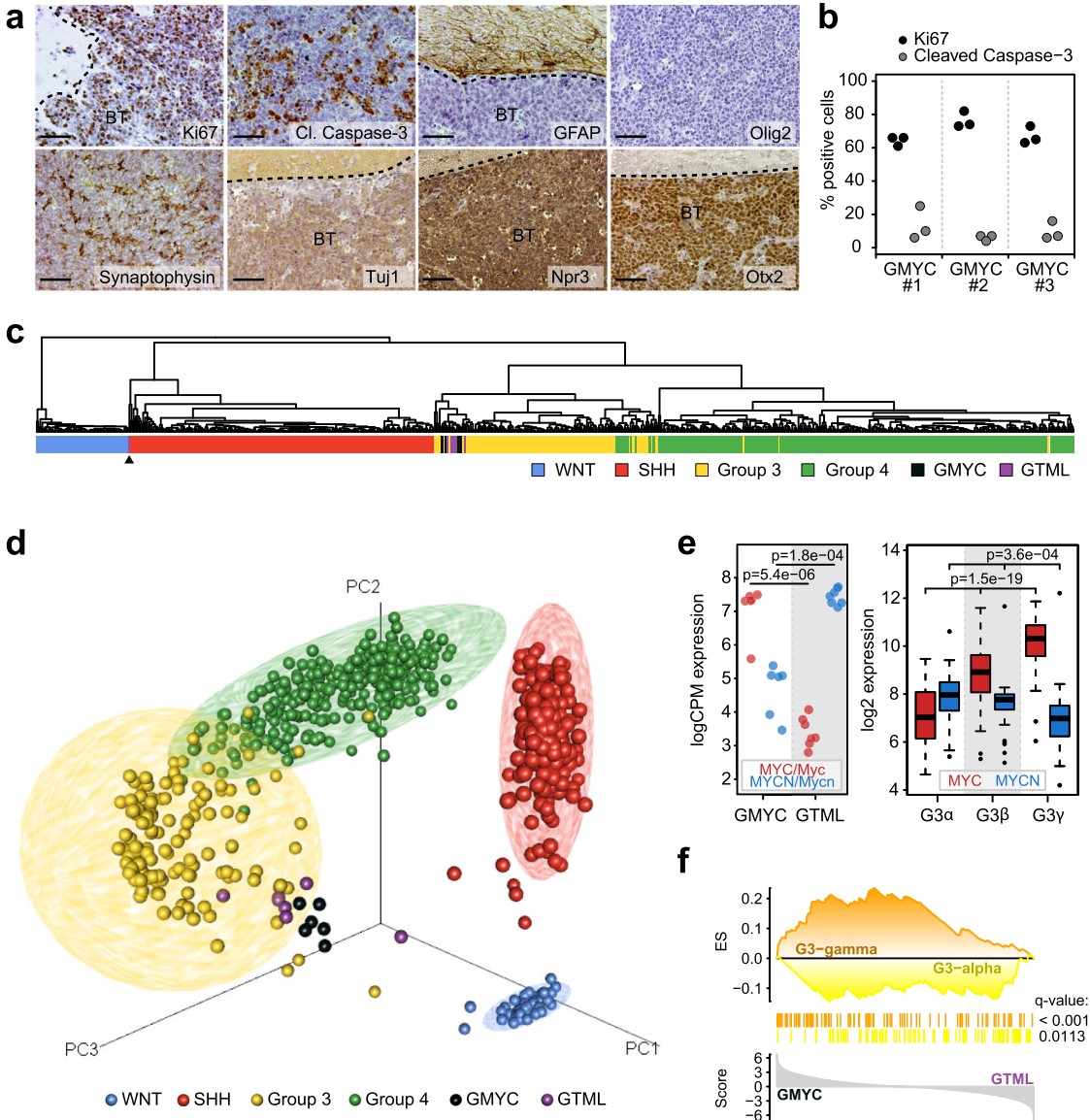

**Fig. 2 | GMYC tumors resemble distinct human MYC-driven Group 3 MB subtypes. a** ×20 magnification. Immunostaining for Ki67 shows a proliferative tumor with pockets of apoptotic cells (Cleaved caspase-3). Brain tumors (BT) stained mostly negative for GFAP and Olig2, had moderate levels of Synaptophysin, TuJ1, and Npr3, and strong Otx2 positivity confirming similarity to a non-SHH MB-like tumor. Scale bar represents 100 µM. **b** Quantification of Ki67 and Cleaved Caspase-3 activity from three representative micrographs in three individual tumors. **c** Hierarchical clustering following cross-species projection of GTML (*n* = 8) and GMYC (*n* = 6) tumors onto human MB samples (*n* = 737; GSE85217). The black arrow indicates a single GTML sample that clustered with the SHH subgroup. **d** PCA plot following cross-species projection of GTML (*n* = 8) and GMYC (*n* = 6) tumor models onto human MB samples (*n* = 737; GSE85217). **e** *Myc/MYC* and *Mycn/MYCN* expression levels in GMYC (*n* = 6) and GTML (*n* = 7) models (left) and in MB Group 3 alpha (*n* = 67), beta (*n* = 37) and gamma (*n* = 40) subsets (right). *P*-values indicate the results of a two-sided Welch's t-test (left) or a one-way ANOVA (right). **f** Result of a targeted GSEA comparing GMYC (*n* = 6) and GTML (*n* = 7) tumors against gene sets consisting of genes upregulated in respectively G3-gamma or G3-alpha human Group 3 MBs (from GSE85217).All experimental data from immunostaining was verified in at least two independent experiments. SA: Senescence-associated.

including glial or mixed glioneuronal tumors (e.g. pediatric glioma or brainstem tumors, Supplementary Fig. 1a), other non-glial tumors (e.g. neurofibroma or choroid plexus tumors, Fig. S1B) and embryonal tumors (e.g. CNS-PNET tumors, ETMR or ATRTs, Supplementary Fig. 1c) the mouse tumors always closely resembled medulloblastoma. In a subsequent classification against a large cohort of MB samples (*n* = 737)[33], GMYC and GTML samples demonstrated a high degree of similarity and a clear resemblance specifically to Group 3 MBs (Fig. 2c, d), with the exception of one outlier GTML tumor sample that also revealed a SHH-like signature; and such tumors can occasionally arise in the GTML model as previously shown[17]. Furthermore, the particular

relationship of the *MYC* genes in these models was also highly reminiscent of the *MYC* and *MYCN* expression patterns in Group 3γ and Group 3α MBs, respectively (Fig. 2e). In fact, based on a GSEA analysis utilizing gene sets differentially expressed between Group 3γ and Group 3α MBs, GMYC displayed a distinct association with the Group 3γ signature, while the GTML profiles were more enriched for the Group 3α signature (Fig. 2f). On the other hand, when performing an ssGSEA enrichment analysis of the GMYC and GTML tumors against the human MB subtypes, it is clear that GMYC tumors again model Group 3γ while GTML tumors align more with Group 3α but also with Group 4 tumors (Supplementary Fig. 1d).

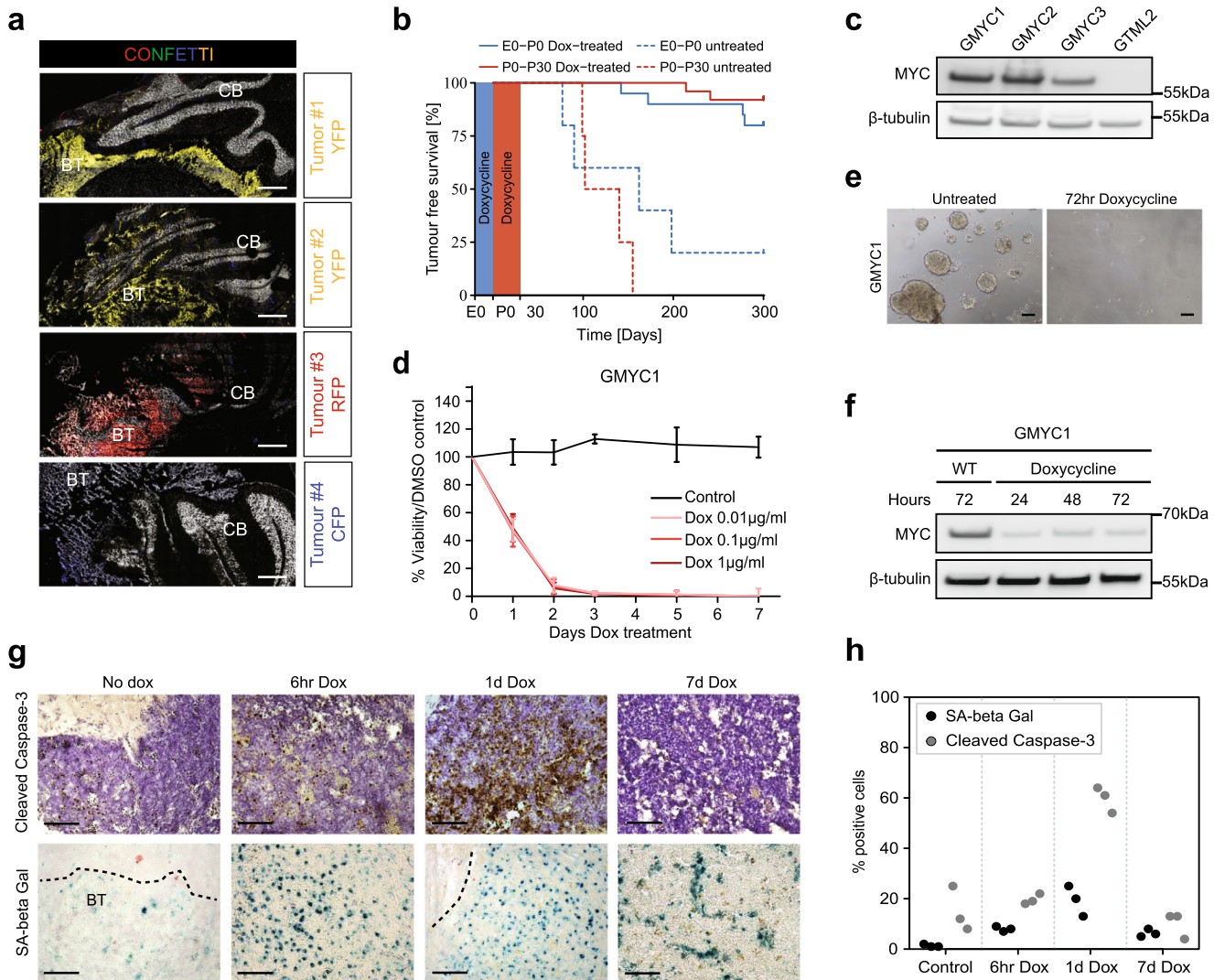

**Fig. 3 | GMYC tumors arise clonally during embryonal development and can be maintained in vitro. a** ×20 magnification. GMYC/Confetti strain of mice suggested that developing tumors (*n* = 4) were of monoclonal origin. Scale bars represent 200 μM. **b** Survival plot of GMYC mice at different windows of embryonic and postnatal dox-treatment. Red line indicates mice who received dox treatment from P0 to P30. Dashed red line is untreated controls from the same breeding. Blue line indicates mice who received dox treatment from embryonic exposure during the mother's gestation until birth (P0). Dashed blue line is untreated controls from the same breeding. **c** Protein analysis of lysates taken from GMYC cell lines expanded in vitro show high levels of MYC expression. Cells from the GTML model do not overexpress MYC protein. **d** GMYC cells can be dox-treated in vitro and are rapidly ablated following exposure. *n* = 3 for each treatment variable. Mean ± SD. **e** Micrograph of GMYC1 cells growing in vitro, both untreated and treated for 72 hours with dox. Scale bars represent 100 μM. **f** Protein analysis of dox-treated GMYC1 cells in vitro. MYC levels are rapidly diminished upon dox (1 μg/ml) treatment. **g** In vivo dox-treatment reveals that tumors rapidly undergo apoptosis following loss of MYC. A proportion of cells are pushed into a senescent state. This response peaks following 24 h of dox-treatment. Scale bars represent 100 μM. **h** Quantification of cleaved caspase-3 and SA-β Gal activity from three representative micrographs as seen in **g**. Blots in **c** and **f** are representative data from a minimum of three biological replicates. All experimental data from treatments (**d**, **e**, **g**) were verified from at least two independent biological replicates. CB cerebellum, BT brain tumor.

## GMYC tumors arise clonally during embryonal development and can be maintained in vitro

Whether human malignant brain tumors arise as a consequence of monoclonal development with accumulation of mutations/alterations, or if they arise from multiple cells of origin in parallel, is difficult to check but heavily debated[34]. More is known about brain cancer evolution where it is clear that MBs generate subclones during progression and that mutations generated during primary tumorigenesis differ substantially from those found at recurrence when studying matched patient sample sets[35]. The multicolor fluorescent reporter system, Confetti[36], allows lineage tracing of cells and can reveal if sporadic tumors in transgenic animals originate from one or from multiple cells. We studied tumor cell populations in a number of mice where the GMYC strain was crossed together with the R26-Confetti strain. An event of CRE-recombination gives rise to expression of one of four possible fluorescent proteins at the same time as the induction of overexpression of *MYC*. All possible fluorophores were seen throughout hindbrain cells expressing the Glt1 promoter; however, in the brains analyzed, there was a preponderance for tumor cells expressing red fluorescent protein (RFP) and yellow fluorescent protein (YFP). Six GMYC/Confetti tumors were visualized to determine clonality (Fig. 3a). Out of these, 2 were negative and 4 showed clear fluorescent signal in the tumor mass. All tumors analyzed were almost exclusively monochromatic, which suggests that the final tumor population is monoclonal and arises from a single, dominant clone.

To investigate the effects of dox-induced *MYC* repression in early brain development, mice were exposed to dox through maternal breast-feeding or through dox-supplemented food from P0 to P30.

Inactivation of the TRE promoter and hence prevention of activation of *MYC* in embryonic GMYC mice led to prevention of tumor formation and a complete tumor-free life span in the majority (84%) of cases (Fig. 3b). GMYC mice that were exposed to dox during the gestation period from E0-E21/P0 saw similar inactivation of *MYC* and a subsequent tumor-free life span in 90% of cases, suggesting that these tumors arise during embryonic development.

GMYC tumor cells could be isolated and maintained in vitro as neurospheres in stem-like conditions. Cultured cells retained high levels of *MYC* expression (Fig. 3c). Neurospheres exposed to dox underwent cell death with no further proliferation (Fig. 3d, e), suggesting that the established cell cultures retained their sensitivity to dox. GMYC cells (GMYC1-3) treated with dox over a period of 3 days died with distinct reduction in total MYC protein (Fig. 3f and Supplementary Fig. 2a–f). Analysis of brain and tumor tissue taken from dox-treated mice after differing periods of treatment showed that withdrawal of *MYC* expression and subsequent tumor ablation is a result of tumor cells rapidly undergoing apoptosis as well as senescence, where both phenomena peak following 1 day of dox-treatment (Fig. 3g, h). As previously mentioned, withdrawal of MYC expression in tumor-presenting mice via dox treatment led to complete regression of the tumor. This regression can be followed via luciferase expression and presents an exciting opportunity for monitoring both in vitro and in vivo treatments (Supplementary Fig. 2g). Cultured GMYC1 cells retained their tumorigenic potential and generated secondary, allografted tumors when injected into the cerebellum of immunocompetent pups and adult mice. Tumors were phenotypically visible at an average of 18 days post-injection (Supplementary Fig. 2h, i). Allografted tumors demonstrated the same staining pattern as the primary tumors and retained high levels of *MYC* (Supplementary Fig. 2j).

## MYC and MYCN-driven tumors have distinct profiles with low ARF levels in GMYC MBs

While both GMYC and GMTL models closely affiliated with Group 3 MBs, RNA-seq analysis illustrated that each model was driven by the expression of its own *MYC* gene, i.e. *MYC* in the GMYC model and *MYCN* in the GTML model (Fig. 2e). Therefore, we asked whether these two models might recapitulate different compartments within Group 3 MBs, of which about 10-17% display *MYC* amplifications and 2–4% harbor *MYCN* amplifications[28,37]. Utilizing methylation-derived copy number intensities and gene expression data (GSE85212 and GSE8217), we identified 24 and 7 human Group 3 MB samples with putative amplifications in *MYC* or *MYCN*, respectively, (Supplementary Fig. S3a, b), which also overlapped with the cases displaying the highest expression of *MYC* (MYC-high, $n = 10$) or *MYCN* (MYCN-high, $n = 3$) in that subgroup (Supplementary Fig. 3c).

Comparing the transcriptional profiles between the latter two categories, we established a signature of genes differentially upregulated in the *MYC*-high ($n = 58$) and *MYCN*-high ($n = 36$) Group 3 MBs, respectively (Supplementary Fig. 3d). Comparing the differential gene expression between GMYC and GTML against these two signatures via a preranked GSEA analysis, we found that GMYC and GTML demonstrate closer affiliation with the human Group 3 *MYC*-high and *MYCN*-high subsets, respectively (Fig. 4a). To further characterize and delineate the two tumor models, we performed an unbiased GSEA. The top significantly enriched gene sets indicated that GMYC tumors exhibited stronger MYC downstream signaling, and upregulation of mTOR and oxidative phosphorylation markers, while GTML tumors exhibited upregulation of genes involved in sensory perception, phototransduction as well as cilium movement (Fig. 4b).

Consistent with the previously identified associations of the models with Group 3 subsets (Fig. 2f) and the distinct photoreceptor signature of Group 3 MBs[38], the observed differential enrichment of the photoreceptor gene set between the models was well recapitulated by corresponding differences between Group 3α and Group 3γ MBs

(Fig. 4c), suggesting again that GTML tumors might be more similar to Group 3α MBs. Furthermore, in Group 3γ MBs the enrichment of the photoreceptor gene set appeared negatively correlated with *MYC* suggesting that photoreceptor activity could be coupled to relative expression of *MYC* and *MYCN* (Supplementary Fig. 3e).

Further comparison of the tumor models via a differential gene expression analysis also revealed that *Cdkn2a* and *Cdkn2b* were both among the top 10 significantly upregulated genes in GTML as compared to GMYC (Fig. 4d), suggesting putatively different roles of these tumor suppressors, including *ARF*, in MYC- and MYCN-driven Group 3 MBs.

While there were potentially too few samples to compare the expression of the human orthologs between *MYC*-high and *MYCN*-high samples in Group 3 human MBs (Supplementary Fig. 3f), we found significant upregulation of *CDKN2A* in *MYCN*-high patients against *MYC*-high samples when including patients from both Group 3 and Group 4 (Fig. 4e and Supplementary Fig. 3g). Generally, *CDKN2A* was significantly downregulated in SHH and Group 3 as compared to Group 4 tumors (Fig. 4f)[33]. However, while *CDKN2A* expression levels in SHH MB did not convey any significant survival difference, high expression of *CDKN2A* (but not *CDKN2B*) in Group 3 and Group 4 tumors was associated with a significant increase in average survival time (Fig. 4g). In summary, these findings suggest that the upregulation of *Cdkn2a* in GTML as compared to GMYC might reflect a clinically relevant feature also present and affecting tumor prognosis between *MYC*-high and *MYCN*-high human Group 3/4 MBs.

Previous reports revealed that the GTML model often harbors mutations in the tumor suppressor gene *Trp53*[39]. We therefore sequenced *Trp53* in GMYC tumors and GTML tumors and saw no discernible difference in mutational status between both groups. Here, 5 out of 12 (42%) GTML tumors and 10 out of 24 (42%) GMYC tumors had p53 mutations but it is clear that in half of the samples found with mutations allele frequencies are less than 50% (Supplementary Table 1). Thus, other key events or co-drivers could still be required in the MYC-dependent GMYC model and in *MYC*-amplified patient tumors.

## Loss of ARF increases tumorigenicity, metastasis, and promotes HGG-driven malignancy

We next investigated if there might be a different usage of *Arf* and *Ink4a* between GMYC and GTML tumors. Indeed, while GMYC samples displayed almost no RNA-seq read counts for either transcript, read counts in the GTML samples seemed to indicate a predominant usage of the *Arf* transcript rather than the *Ink4a* transcript (Fig. 5a). In turn, the observed differential expression of *Cdnk2a* between GMYC and GTML can to a large part be contributed to differential expression of the *Arf* transcript (Fig. 5b). Furthermore, a comparison between Group 3 patients with putative *MYC* or *MYCN* amplifications indicated a significant differential promoter methylation of *ARF* with frequent methylations in the *MYC*-amplified group and hypomethylation in the *MYCN*-amplified tumors (Supplementary Fig. 4a). However, when comparing the expression levels of *CDKN2A* of hyper- and hypomethylated human samples we could not find any significant difference (Supplementary Fig. 4b). This might suggest that methylations found at the *ARF* promoter are not leading to a decrease in *ARF* gene expression, or that there is a compensatory increase of *INK4A* gene expression in samples in where *ARF* might be suppressed, as readily shown before[40]. Unfortunately, however, there is no public data available to study such compensation for these sets of tumors. We similarly checked genomic methylation status in our GMYC and GTML models using methylation profiling with the MM285 Infinium Mouse Methylation BeadChip. We identified common epigenetic silencing in the CDKN2A locus in about half of the samples and again methylation upstream of the *Arf* transcriptional start site in the tumors (Supplementary Fig. 4c). However, there was no significant

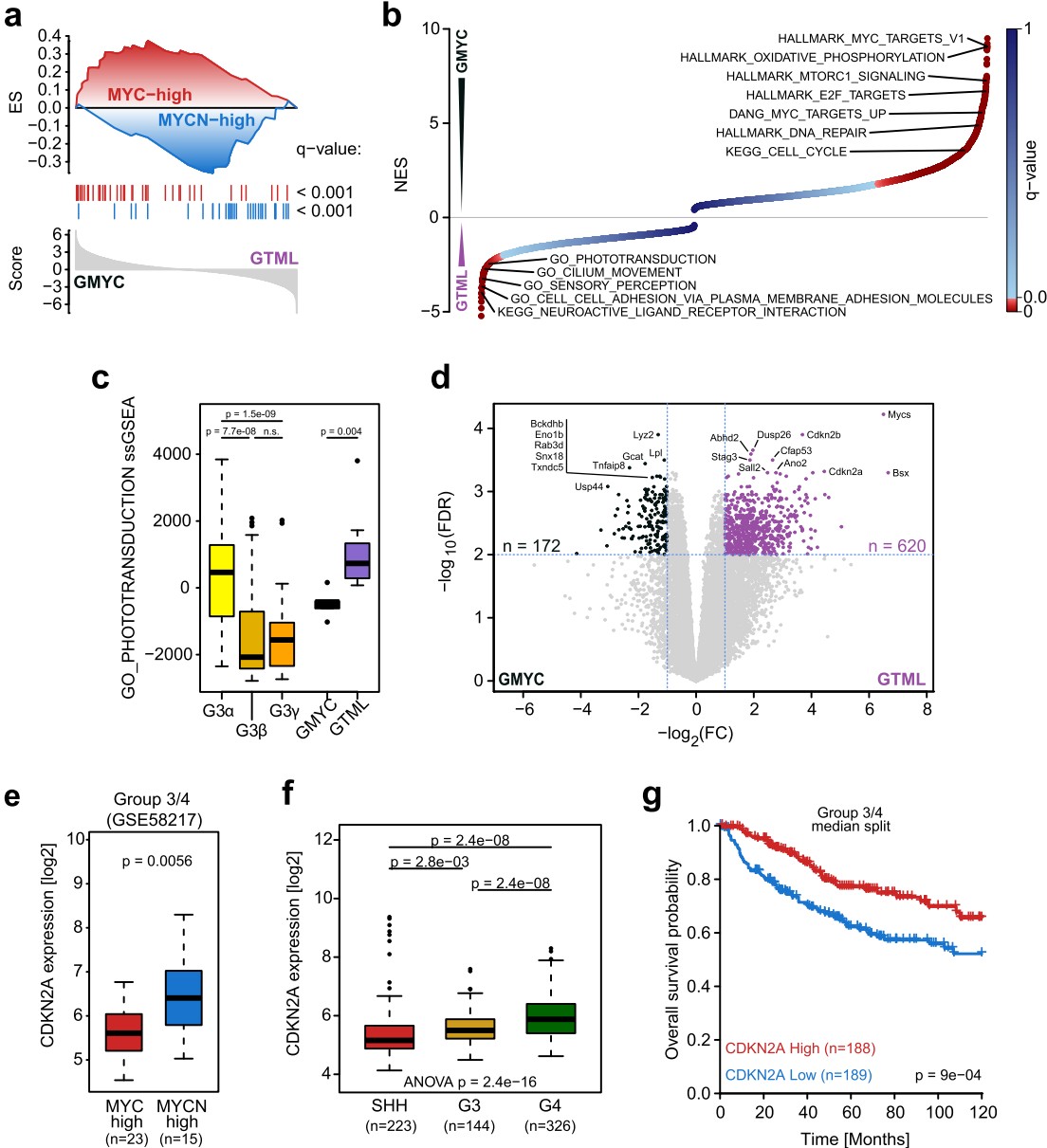

**Fig. 4 | MYC and MYCN-driven tumors have distinct profiles with low ARF levels in GMYC MBs. a** Results of a targeted GSEA comparing the differential expression of genes between GMYC ($n = 6$) and GTML ($n = 7$) tumors against gene sets consisting of genes upregulated in respectively MYC-high or MYCN-high human Group 3 MBs (from Supplementary Fig. 3d). **b** Illustration of normalized enrichment scores (NES) obtained from an unbiased GSEA comparing GMYC ($n = 6$) and GTML ($n = 7$) tumors. Significantly enriched gene sets for GMYC (NES > 0) and GTML (NES < 0) are indicated as red dots. **c** Boxplot depicting ssGSEA enrichment scores for the phototransduction gene set identified in **b** between subsets of human Group 3 MBs (from GSE85217), GMYC ($n = 6$), and GTML ($n = 7$) models. *P*-values indicate the results of two-sided Wilcoxon Mann-Whitney rank sum tests (n.s.: $p > 0.05$). **d** Volcano plot depicting the differential expression between the 7 GTML and 6 GMYC tumor samples classified as Group 3 MB. The 10 top significantly (as measured by lowest FDR-values) genes for each model are highlighted. The horizontal dashed line indicates the FDR = 0.01 threshold, while the vertical dashed lines indicate a logFC of -1 or 1, respectively. The *Myc* and *Mycn* genes were removed from the ranked gene list. **e** Box plot comparing the expression of *CDKN2A* between MYC-high and MYCN-high Group 3/4 samples (see also Supplementary Fig. 3g). *P*-value was computed using two-sided Welch's *t*-test. **f** Box plot comparing the expression of *CDKN2A* between SHH, Group 3, and Group 4 MB samples. *P*-value was computed using two-sided Welch's *t*-test. **g** Kaplan–Meier plots comparing the survival of patients with low and high gene expression of *CDKN2A* within the Group 3 or Group 4 subsets (from GSE85217). Within each subset, the sample groups were established by separating samples based on the median expression of *CDKN2A*. Log-rank Mantel-Cox statistical test.

difference in methylation patterns when comparing GMYC and GTML tumors despite the fact that the *Arf* mRNA levels are different (Fig. 5b).

Having identified the differential expression of *Arf* in both models, we investigated whether modification of the *Cdkn2a* gene, encoding p19ARF, is involved in tumor formation in the GMYC model. We crossed the GMYC and GTML strains with a TRE-CRE strain[31] and an heterozygous or homozygous *Arf* (but not Ink4a) gene-specific floxed strain[41] in order to generate partial and complete *Arf* knockout strains of each TRE-driven model.

Considering that the baseline GMYC tumor penetrance and latency were 60% and an average of 130 days, respectively, partial knockout of *Arf* in this model led to no change in either parameter. However, complete knockout of *Arf* in the GMYC model increased tumor penetrance to 95% and decreased latency to an average of 100 days. Partial knockout in the GTML model increased tumor

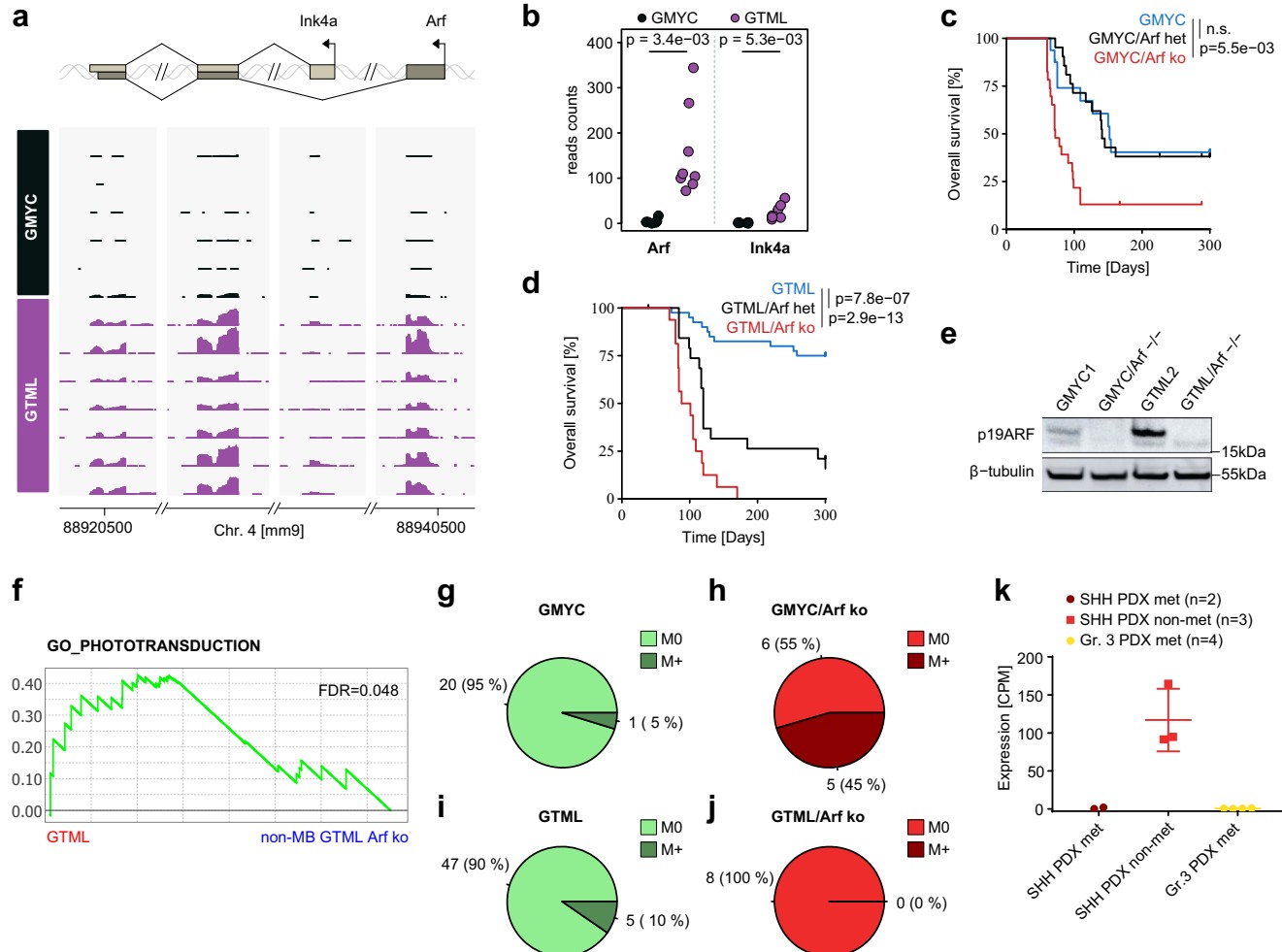

**Fig. 5 | Loss of ARF increases tumorigenicity, metastasis, and promotes HGG-driven malignancy. a** Schematic display of transcriptional expression (of indicated exons) for *Arf* and *Ink4a* transcripts (from RNA-Seq) aligned to the CDKN2A locus on mouse Chromosome 4 between individual GMYC (*n* = 6) and GTML (*n* = 7) tumors. **b** Strip chart comparing the read counts for *Arf* or *Ink4a*, respectively, between GMYC and GTML tumors. *P*-values indicate the results of a two-sided Welch's t-test. **c-d** Kaplan-Meier plots comparing the subsequent changes in penetrance and survival following heterozygous and homozygous loss of *Arf* in the GMYC and GTML models. Heterozygous loss had no effect on overall survival for GMYC mice (*p* > 0.05), whereas homozygous loss dramatically increased penetrance and reduced overall survival (*p* = 5.5e-03). In the GTML model, heterozygous loss increased tumor penetrance and reduced overall survival (*p* = 7.8e-07), and homozygous loss impacted overall survival further (*p* = 2.9e-13). Log-rank Mantel-Cox statistical test. **e** Protein analysis of p19ARF in GMYC and GTML cell lines established from in vivo material, then grown in vitro. Following complete

knockout of the *Arf* gene there is subsequent total reduction of protein product. **f** GSEA analysis of GO_PHOTOTRANSDUCTION with a significant enrichment in GTML cells (MB-like) as compared non-MB GTML cells. Enrichments were considered significant if FDR < 0.05. **g–j** Comparison of metastatic events observed in the GMYC and GTML models upon complete knockout of *Arf*. GMYC model had relatively few events of metastasis, 1 in 21 (5%). However, with homozygous loss of *Arf*, many mice harbor metastasis, 5 in 11 (45%). In GTML metastasis was observed for 5 out of 52 tumors (10%). This did not increase upon homozygous loss of *Arf*, 0 in 8(0%). **k** *CDKN2A* expression (RNA-seq) levels in individual MB PDXs for SHH met (*n* = 2), SHH non-met (*n* = 3) and Gr. 3 met (*n* = 4) medulloblastoma subgroups (from GSE106728) show that heavily suppressed CDKN2A levels coincide with metastatic dissemination. Scatter dot plot presented as mean values ± SD. Experimental data from immunostainings (**e**) was verified in at least two independent biological replicates.

penetrance from 40% to 90%, and latency decreased from 200 days to 150 days. Complete knockout of *Arf* in the GTML model increased tumor penetrance to 100% and further decreased latency to 80 days (Fig. 5c, d). The effects of gene knockout on protein expression of isolated tumor biopsies can be seen in Supplementary Fig. 5a where *Arf* is totally depleted in *Arf*−/− tumors and partially suppressed in *Arf*+/− tumors as compared to *Arf*+/+ tumors. It was further evident that *Arf* was silenced also in some *Arf*+/+ tumors. Cell lines established from GTML tumors exhibited high baseline levels of ARF, while cell lines from GMYC tumors exhibited low baseline levels of ARF; following complete knockout of *Arf* there were no detectable levels of ARF in cultured cells of isolated tumors from the different crosses (Fig. 5e). Strikingly, while some *Arf*-depleted tumors were still Group 3-like, complete *Arf* depletion led to an increased ratio (4 out of 6 tumors) of

glial, HGG-like brain tumors in both GTML and GMYC tumors. These non-MB GTML Arf ko and non-MB GMYC Arf ko tumors were most similar to HGG-G34 or HGG-RTKs as shown in tSNE plots when performing cross species RNA-seq analysis against specific HGG subtypes and ependymoma (Supplementary Fig. 5b–d). They showed increased expression of differentially expressed forebrain tumor genes found in both HGG-G34 tumors[42] and markers that define a core set of glioma propagation transcription factors, such as *Pou3f2*, *Sall2*, *Sox2*, and *Olig2*[43], and an upregulation of nestin and the platelet-derived growth factor (PDGF) pathway commonly seen in HGG-RTKs[20,44] (Supplementary Fig. 5e). These tumors had further lost Group 3 MB specific NPR3 and OTX2 expression (Supplementary Fig. 5f). For GTML tumors that have the highest levels of photoreceptor pathway activation (Fig. 4c), *Arf* depletion led to a significant reduced association with

GO_PHOTOTRANSDUCTION in non-MB GTML Arf ko tumors that developed into HGGs (Fig. 5f). It has recently been shown that OTX2 is a master regulator of the photoreceptor pathway and that its suppression in Gr.3 MB lines showed a significant enrichment in regulation of neuron projection development as well as a reduced ribosome biogenesis gene set activation[6]. Intriguingly, these gene sets were similarly differentially regulated in our GSEA of non-MB GTML Arf ko tumors showing significant enrichment in regulation of neuron projection development as compared to MB GTML tumors in were ribosome biogenesis was enriched (Supplementary Fig. 5g). When checking MYC pathway activity in *Arf*-depleted tumors (including tumors resembling both MB and non-MB entities) we surprisingly found a significantly reduced association with MYC as well as MYCN gene sets[45] as compared to *Arf* wild-type GMYC and GTML tumors (Supplementary Fig. 5h). When checking cell viability following 72 h of dox treatment, it was also evident that *Arf*-depleted GMYC cells were still dependent but significantly more resistant to dox treatment and subsequent MYC suppression as compared to GMYC1 cells. This suggests that *Arf*-deficient mouse tumors have a decreased dependence on the MYC oncogene (Supplementary Fig. 5i).

As previously mentioned, spinal cord metastasis was not a common event in our GMYC model (observed in only 5%). Complete knockout of *Arf* increased this event, where 5 out of 11 (45%) spinal cords showed leptomeningeal dissemination (Fig. 5g, h). MYCN-driven GTML tumors showed very low metastatic events (10%) and this frequency did not increase with knockout of *Arf* (Fig. 5i, j) but on the other hand, most of these tumors resembled HGGs that rarely metastasize to the spinal cord in patients. Similarly, this was also observed when looking at data comparing the frequency of metastatic dissemination and extent of *CDKN2A* expression in a set of seven SHH and Gr.3 PDX models - low levels of *CDKN2A* expression coincided with PDXs that generated spinal cord metastases following stereotactic injection into the cerebellum (Fig. 5k). PDXs used within this study were previously established PDX lines of four SHH MBs (ICb-984MB, BT-084, DMB012, RCMB18) and three Gr.3 MBs (ICb-1572MB, Med-411-FH, RCMB40), as previously described[16].

## MYC but not MYCN promotes ARF suppression, which could be restored by DNMT inhibition

We next developed a regulatable model to in detail explore the de novo effects of MYC on previously formed GTML tumor cells in vitro. GTML cells (GTML2) were transduced with a lentiviral construct overexpressing *MYC*. MYC was absent in the GTML2 cells but was elevated upon MYC lentiviral overexpression, coinciding with a reduction of MYCN levels and an increase of INK4A protein levels. Comparison of apoptosis showed a greater response in the GTML2/+MYC cells, possibly a result of MYC pushing more of these cells into an apoptotic state compared to MYCN (Fig. 6a). However, when GTML2/+MYC cells were treated with dox leading to MYCN suppression, there was an initial response of cell death which reached a peak of 40% viable cells. Still, *MYC* was able to rescue these MYCN-suppressed cells and became the key driver of proliferation, leading to complete rescue and survival of tumor cells (Fig. 6b).

We next wanted to investigate whether MYC, but not MYCN is responsible for de novo suppression of ARF in transformed cells. As shown previously, ARF levels were higher in GTML2 cells than in GMYC1 cells, and ARF levels remained high in the untreated GTML2/+MYC line (Supplementary Fig. 6a). Furthermore, when MYCN was suppressed in GTML2/+MYC cells via dox treatment, there was a reduction in total ARF protein by day 7, indicating that addition of MYC was either promoting *CDKN2A* suppression or a selection of surviving cells with lower CDKN2A levels, and this temporal suppression was reversed when MYCN levels were restored following 7 as well as 21-day restoration (Fig. 6c and Supplementary Fig. 6b). In a similar manner, we next examined if MIZ1 was required for suppression of ARF in these

MYC-driven tumors as has been previously reported when MYC recruits MAX and MIZ1 to suppress CDKN2B in cells[46]. By using a MYC mutant construct (MYCV394D aka. MYCVD) that is unable to bind MIZ1[47], we studied if ARF suppression was disabled when using the MYCVD mutant instead of the wild-type MYC vector. However, lentivirally transduced and overexpressed MYCVD was still able to efficiently suppress ARF levels in GTML2 cells (Fig. 6d and Supplementary Fig. 6c). This suggests that MIZ1 is likely not required for the MYC-driven suppression of ARF levels.

Given that methylation of the *Cdkn2a* locus was commonly found in the mouse tumors (Supplementary Fig. 4c) and to test if *Arf* expression could be restored by demethylation, we treated GMYC and GTML cells with the DNMT inhibitor 5-Azacytidine in vitro. 5-azacytidine treatment resulted in an increase of ARF expression in the GMYC1 cells but not in the GTML2 cells. In contrast, 5-Azacytidine had no effect on ARF levels in GTML2 cells (Fig. 6e). A dose-response curve was generated in order to determine EC50 values of 5-Azacytidine in GMYC1 and GTML2 cells – this was calculated to be approximately $0.5\,\mu M$ for both lines and there was no significant difference in effect on cell survival for both cell lines (Fig. 6f, g). To determine if restoration of *Arf* led to stabilization of p53 and subsequent cell death, we cultured GTML and GMYC cells in the presence of 5-Azacytidine and monitored their proliferation and viability over 3 days. 5-Azacytidine treatment had an effect on overall survival in GMYC cells but also in GTML cells, but only at the highest concentration ($1\,\mu M$) used (Fig. 6h and Supplementary Fig. 6d) suggesting cells are affected by global demethylation effects.

To further investigate if demethylation could work as part of a combinatorial treatment, GMYC1 and GTML2 cells were treated with 5-Azacytidine and cisplatin, an alkylating agent commonly used in MB treatment. Individual treatments reduced overall survival and proliferation of cells. However, the treatment combining both 5-Azacytidine and cisplatin did not significantly reduce survival of cells further (Fig. 6i and Supplementary Fig. 6e), indicating a doubtful role for 5-Azacytidine when used in combination with standard chemotherapy.

## HSP90 inhibition suppresses MYC-driven tumors in an ARF-dependent way

Utilizing the results of our previous GSEA analyses, we identified a handful of pharmaceutical agents that were predicted to significantly elicit a greater response in GMYC as compared to GTML and, when compared to human MB, also suggested a better treatment response in human Group 3γ compared to Group 3α and Group 4 MB (Fig. 7a). As anticipated based on the GSEA enrichment for the mTOR pathway in GMYC tumors (see Fig. 4b), the mTOR inhibitor rapamycin was predicted to have a significant response in Group 3γ compared to Group 3α and Group 4 MB. However, as rapamycin and more specific mTOR inhibitors have already been tested and shown great response in our GTML model[48] as well as in our recent MYCN-driven humanized MB SHH tumors[16], we were more focused on finding a compound that would be particularly sensitive in MYC-driven rather than MYCN-driven MB. Next on the list were response to RAS inhibitors, like Salirasib. Although *RAS* genes show a correlation to metastatic medulloblastoma[49] they are not found significantly upregulated or amplified in Group 3 MBs[28]. Further, GSK3-beta inhibitors like SB216763 that was found as a top hit has been tested in Group 3 medulloblastoma before but only shown modest cytotoxic efficacy alone[50].

Instead, the expression of *HSP90AB1*, an important member of the heat shock protein 90 (HSP90) family (coupled to the MALONEY_RESPONSE_TO_17AAG_DN gene set in Fig. 7a), appeared significantly associated with patient survival (Fig. 7b) and highly correlated with *MYC* expression (Fig. 7c). HSP90 inhibitors activate the heat shock factor 1 (HSF1), proteotoxic stress response pathway[51].

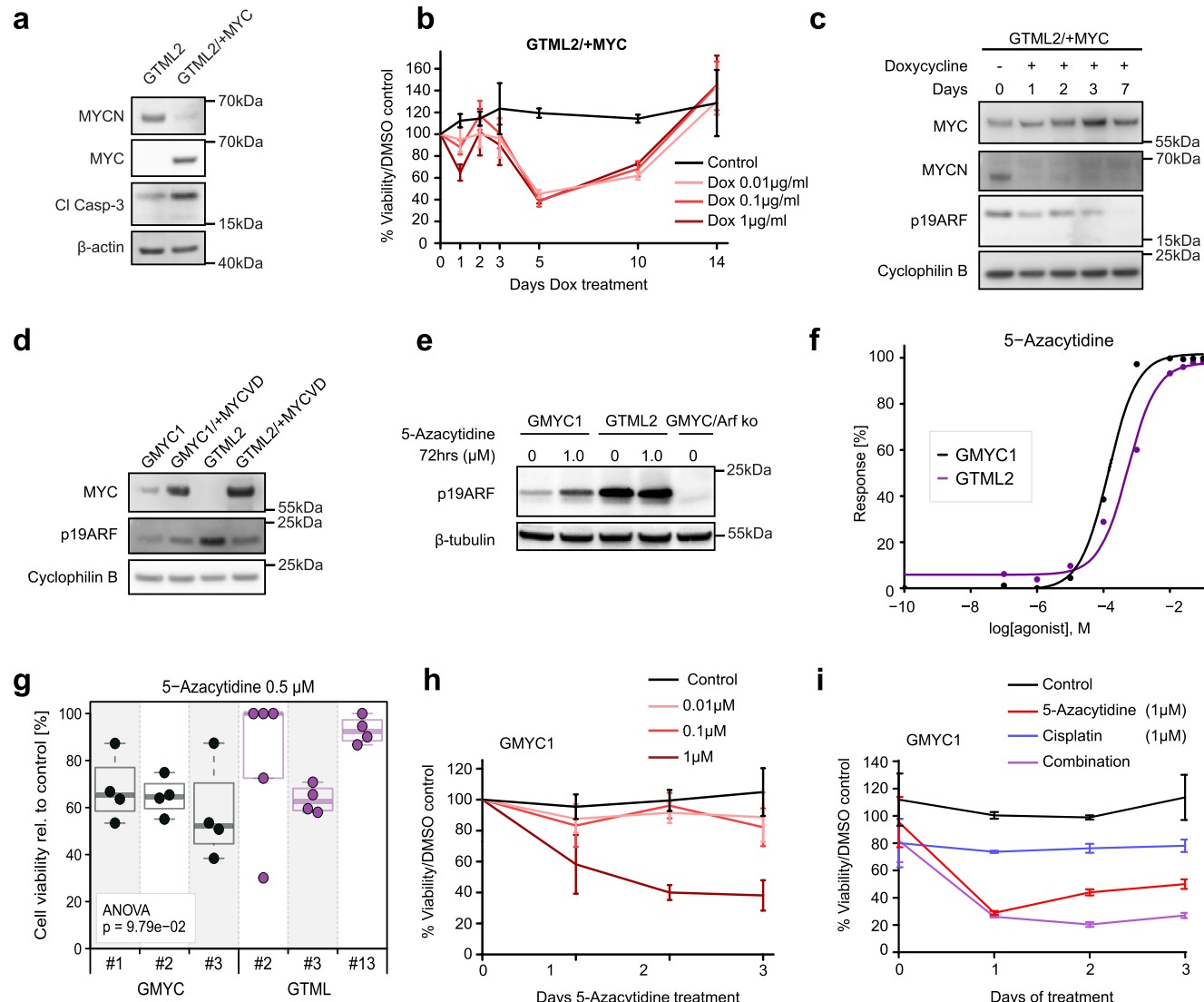

**Fig. 6 | MYC but not MYCN promotes ARF suppression, which could be restored by DNMT inhibition. a** Protein analysis of GTML2 and GTML2/ + MYC cell lines. MYCN expression and apoptotic activity decreases following dox treatment when replaced by MYC expression in this system. **b** GTML2/ + MYC cells treated with dox in vitro see a partial decrease in cell survival and viability when MYCN expression is suppressed, but cells are selected and proliferate when MYC likely becomes the new oncogenic driver. *n* = 3 for each treatment variable. Mean ± SD. **c** GTML2/ + MYC cells treated with dox in vitro over a period of 7 days show an increase in MYC levels and reduction of MYCN levels during the dox/TetOFF switch. This switch leads to reduction of ARF levels from de novo suppression of *ARF*. **d** Protein analysis of GMYC1, GMYC/ + MYCVD, GTML2, and GTML2/ + MYC cell lines. **e** Protein analysis of GMYC1 and GTML2 cell lines. Demethylation agent, 5-Azacytidine, shows a concentration gradient of ARF upregulation in response over 12 h of treatment in GMYC1 cells. Such as response is not elicited in GTML2 cells. **f** Dose response curve showing a similar, non-significant difference of 5-Azacytidine treatment on GMYC and GTML cells. **g** GMYC and GTML cells treated with the calculated EC50 value (for GMYC cells) for 5-Azacytidine concentration. ANOVA analysis shows there is no difference in treatment response to this agent for MYC- or MYCN-driven cell lines. **h** GMYC1 cells were treated in vitro with 5-Azacytidine over 3 days. The highest concentration (1 µmol/L) caused a reduction in cell viability and proliferation. *n* = 3 for each treatment variable. Mean ± SD. **i** GMYC1 cells were then treated in vitro with 5-Azacytidine (demethylation), cisplatin (alkylation), or combination treatment over 3 days. Independent treatments saw a reduction in cell viability and proliferation. *n* = 3 for each treatment variable. Mean ± SD. All experimental data from immunostainings (**a, c, d, e**) and treatments (**b, f, h, i**) was verified from at least two independent biological replicates.

Further, there was a significant gene set enrichment of HSF1-mediated heat shock response in MYC high / CDKN2A low Gr. 3 patients compared to patients with MYC low / CDKN2A high levels (Fig. 7d). This enrichment was not seen when similarly comparing MYC high to MYC low or CDKN2A high to CDKN2A low Gr. 3 patients. Combined, this suggests that HSP90 inhibition identified as a top candidate on our list might be a promising route for treating *MYC*-overexpressing MBs. Based on these findings, we thus performed treatments of GMYC and GTML cell lines with Onalespib, a selective and potent HSP90 inhibitor[52]. Dose responses indicated greater sensitivity of GMYC cells to HSP90 inhibition (Fig. 7e) and subsequent and significant cell death

in GMYC cells compared to GTML cells (Fig. 7f). Approximate EC50 values for GMYC1 and GTML2 cells treated with Onalespib were 0.05 µM and 2 µM, respectively. HSP90 inhibition by Onalespib led to increased level of HSF1 and HSP70 in GMYC cells indicating a well-recognized pattern of HSF1-mediated stress response leading to HSP70 induction from successful HSP90 inhibition as previously described[51,53]. Importantly, Onalespib could restore ARF levels in GMYC cells as compared to GTML cells, consequently inducing cell cycle arrest and apoptosis from increasing levels of p21 and Cleaved Caspase-3 (Fig. 7g). To further establish the effectiveness of HSP90 inhibition in combination with conventional treatment modalities,

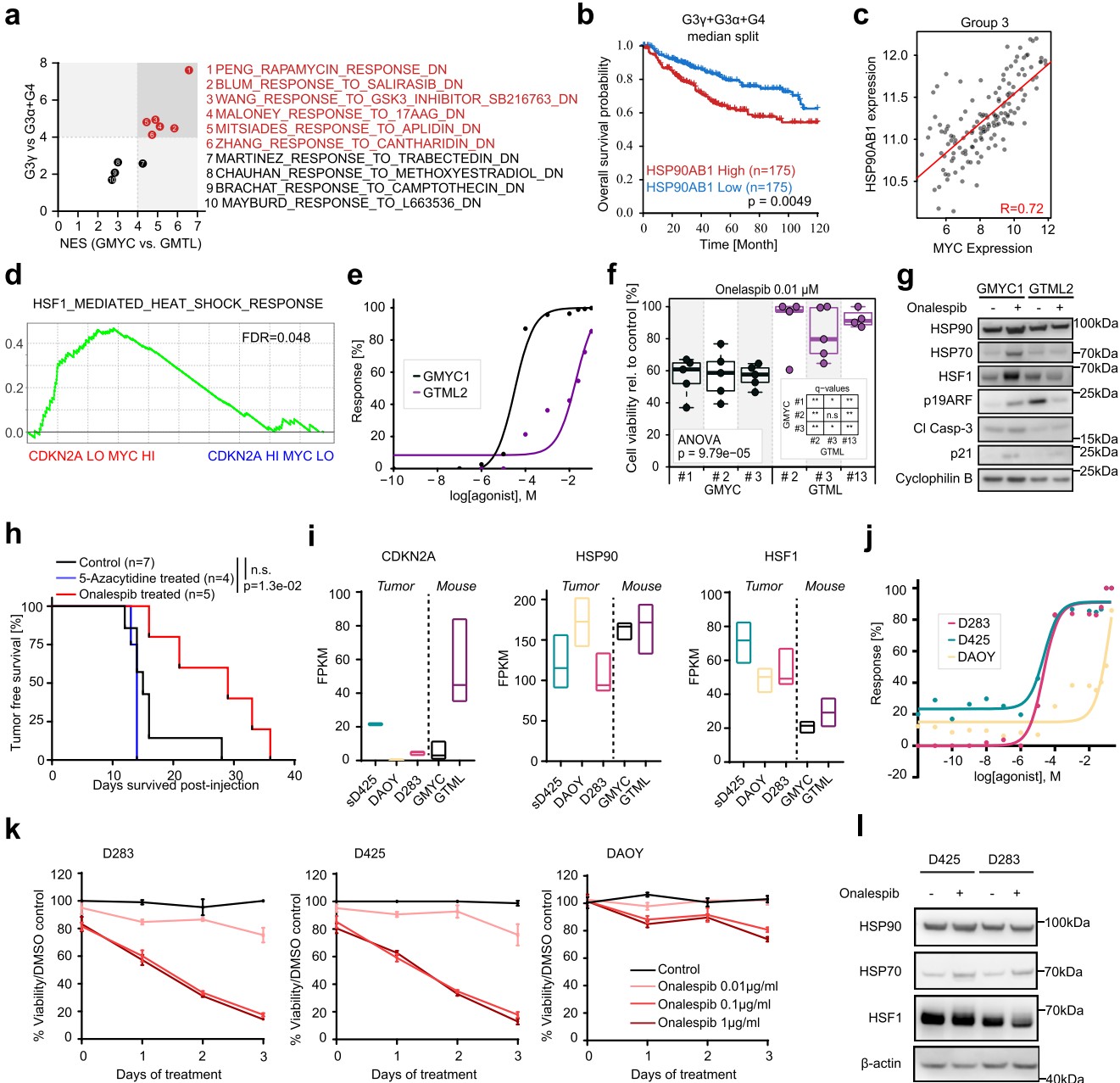

**Fig. 7 | HSP90 inhibition suppresses MYC-driven tumors in an ARF-dependent way. a** Normalized enrichment scores (NES) of 10 drug gene sets between the GMYC-vs-GTML GSEA analysis and G3γ-vs-G3α + G4 GSEA analysis. Gene signatures were identified as the top 10 significant gene sets reported to be repressed by drug treatment and upregulated in GMYC. Lines indicate a threshold of NES = 4 and red highlighted drug gene sets indicate the top candidates with a NES > 4 in both GMYC and G3γ. **b** Kaplan-Meier plot comparing difference (log-rank (Mantel-Haenszel) test) of overall survival between G3γ + G3α + G4 MB patients with high (top 25%) or low (bottom 25%) expression of *HSP90AB1*. **c** Scatter plot comparing expression of *HSP90AB1* with *MYC* in Group 3 MBs. *R*-value = Pearson correlation coefficient. **d** GSEA analysis of REACTOME_REGULATION_OF_HSF1_MEDIATED_HEAT_-SHOCK_RESPONSE with significant enrichment (FDR < 0.05) in MYC high/CDKN2A low Gr. 3 patients. **e** Significant dose response (*p* < 0.05; Student's *t* test) of onalespib-treated GMYC cells compared to GTML cells. **f** GMYC and GTML cells treated with the calculated GMYC1 EC50 value for onalespib concentration. ANOVA analysis shows a significant difference in response between cell lines. *P*-values indicate the results of a one-way ANOVA. **g** Protein analysis of GMYC1 and GTML2

cells treated with approximate EC50 values of onalespib for 24 h. Treated GMYC1 cells show an increase in HSP70, HSF1, ARF, apoptotic activity, and p21 induction compared to their untreated state and as compared to treated GTML2 cells. **h** Onalespib (compared to control or 5-Azacytidine) treatment significantly increased the survival time for mice engrafted with GMYC/TetGFP/Luc cells (Log-rank Mantel-Cox statistical test). **i** Expression analysis (from GSE107405) from tumor biopsies from our GMYC and GTML animal models and patient derived MB cell lines (sD425, DAOY, D283). Floating bars min to max with a median center line. **j** Dose response curve showing a significant response of onalespib-treated *MYC*-driven, CDKN2A-present D283 and sD425 human cells compared to the DAOY cell line which has minimal expression of *MYC* and *CDKN2A*. **k** D283, sD425, and DAOY cell lines treated with onalespib in vitro over 3 days. Mean ± SD. **l** Protein analysis of onalespib-responsive sD425 and D283 cell lines treated with onalespib for 24 hours in vitro. Treated cells show an increase in HSP70 protein levels. Data from immunostainings (**g, l**) and treatments (**e, j, k**) was verified from at least two independent biological replicates.

representative GMYC and GTML cell lines were treated with their calculated EC50 values of Onalespib and 1 μM of cisplatin. Treatment combining both Onalespib and cisplatin showed significant synergism and increased cell death in the GMYC line, indicating a potential clinical use for Onalespib in combination with standard chemotherapy in MYC-driven MB where ARF is silenced, as compared to MYCN-driven MB (Supplementary Fig. 7a-c).

When treating allografted tumor-bearing mice with I.P. injections 20 mg/kg Onalespib daily for 4 days, survival time was significantly extended (median survival time 27 days) ($p = 1.3e-02$) as compared to the control mice which received daily injections of vehicle (median survival time 16 days) (Fig. 7h). Mice treated with 5-Azacytidine had no increase in survival time (median survival time 14 days), and mice exposed to daily 2 Gy irradiation for 5 days saw an increase in survival time, but this was not significant (median survival time 27 days) ($p = 0.0536$) (Fig. 7h and Supplementary Fig. 7d). Furthermore, we established cell lines from Arf-depleted MYC-driven brain tumors. However, a similar treatment scheme as above with I.P. injections 20 mg/kg Onalespib daily for 4 days, did not significantly extend survival of ARF-deficient allografted animals (median survival time 28 days) ($p = 0.7857$) (Supplementary Fig. 7e) indicating the importance of ARF restoration for beneficial HSP90 treatment. RNA transcription of GMYC1 and GTML2 cells treated with Onalespib for 6 hours showed a significant enrichment of gene sets related to apoptosis, MYC targeting and cellular heat response in GMYC versus GTML cells (Supplementary Table 2). It showed an increased expression of p21 (Cdkn1a) after treatment in GMYC1 cells as compared to GTML2 cells indicative that HSP90 inhibition led to stabilization of ARF and thus reduced MDM2-mediated degradation of p21, which resulted in cells being pushed into an apoptotic state (Supplementary Fig. 7f, g).

Finally, using transcriptional data from several patient derived human MB cell lines, we identified those which closely resembled gene expression from our animal models and thus could be used as human analogs to test the effects of Onalespib inhibition on MYC- or MYCN-expressing lines, and the relevance of CDKN2A and HSP90 expression in human tumors in respect to Onalespib inhibitory treatment. sD425, a recurrent Group 3 MB line over-expressing MYC growing in stem cell media; D283, a Group 3 MB line over-expressing MYC; and DAOY, a MYCN-expressing SHH MB line[45]. Here it was seen that the D283 and DAOY human lines had minimal expression of the CDKN2A gene (Fig. 7i), much like our MYC-driven GMYC model, and sD425 had elevated expression of CDKN2A, though not to the extent of expression in GTML tumors. HSP90 expression was observed across all human and mouse tumors, and HSF1 expression was elevated in human lines as compared to the animal tumors (Fig. 7i). If Onalespib treatment requires presence of ARF and MYC, rather than MYCN, for successful tumor cell inhibition, it would be expected that D283 and D425 cells would be more sensitive to HSP90 inhibition than DAOY cells. This was indeed the case, as indicated with the dose response curve evaluating effective Onalespib dose against these cell lines (Fig. 7j). Furthermore, D283 and D425 cells were more sensitive to this inhibitory treatment and saw extensive cell death and reduction in proliferation when treated over a three-day period. In contrast, DAOY cells were largely unaffected from this same treatment (Fig. 7k). Overall confirming that presence of ARF expression, as well as MYC, but not MYCN, are key identifiers of successful tumor cell response and ablation to HSP90 inhibition in both our animal models and in human lines. Lastly, to confirm that HSP90 inhibitory treatment leading to subsequent cell death in the sensitive human cell lines was due to Onalespib and not off-target effects, we once more checked protein levels of HSP90, HSP70, and HSF1. As also observed in the animal lines, HSP90 protein remained unchanged due to inhibition of activity and not degradation of this protein during treatment, where elevations of HSP70 and the heat shock response indicate successful treatment and disruption of the HSP90/HSP70 complex. HSF1 remained largely unchanged but this

is not surprising due to the already ongoing stress response facilitating growth of these tumor cells. (Fig. 7l). This was further confirmed through GSEA analysis of GMYC1 and GTML2 cells treated with Onalespib, where the GMYC1 cells had significant enrichment of the HSF1-mediated heat-shock-response pathway as compared to GTML2 cells and to Arf-depleted GTML cell lines (Supplementary Fig. 7g, h). The Arf-depleted GMYC cell line had no significant difference in HSF1-mediated heat-shock-response pathway when compared to GTML2 cells, again indicating the importance of presence of functional ARF for activation of this response for efficient HSP90 targeting.

## Discussion

Group 3 MB is associated with elevated MYC levels, photoreceptor activity and very poor survival. In the present study we have addressed the key role of MYC in MB and created a model that accurately resembles the biology of aggressive Group 3 tumors. The GMYC model can be easily adapted, granting the opportunity for many avenues of cell tracing and pharmaceutical testing, both in vivo and in vitro. Tumors in this model develop clonally at embryonal/postnatal time points and show dependence on MYC signaling as tumor cells are efficiently eradicated upon MYC inhibition.

Mutation of TP53 is a rare event not typically seen in Group 3 tumors[28] where p53 is likely still functional. Sequencing of Trp53 in the GMYC and GTML model reveals that they have comparable frequency of p53 mutations, occuring in less than half of tumors studied. If tumor development in pediatric patients, similar to cancer in adults, is considered a multi-step process, there must be acquisitions of secondary genetic events to drive tumorigenesis and propagation. Additionally, the upstream p53 regulator gene, CDKN2A, is rarely inactivated or lost in human medulloblastoma[21]. CDKN2A is instead commonly lost in HGG patients and our data suggests that ARF suppression may be an early event in these tumors while a similar mechanism for p53 inactivation occurs in SHH MB patients that harbor p53 mutations. In our MYC-driven model, we have not identified a full depletion but a significant suppression or silencing of CDKN2A that was also observed in Group 3 patients. Transient transcriptional suppression or methylation of the gene appears to be a driver for tumorigenic initiation of Group 3 MB, reducing the capability of functional p53 without p53-specific mutational removal. Silencing or loss of Arf gene or the downstream p53 gene have occasionally been shown to contribute to malignancy[54], metastatic spread to the brain[55] or recurrence of MBs[35,39] in previous studies. Similarly, ARF suppression dramatically increased malignancy, particularly in the GTML strain that had higher ARF levels in developing tumors than the GMYC strain. Complete knockdown of Arf still significantly increased the instances of leptomeningeal dissemination in the GMYC strain. Arf loss increased the development of tumors resembling pediatric high-grade glioma, especially HGG-RTK and HGG-G34 tumor types. Our data suggest that early ARF/p53 inhibition but not late p53 suppression is important for glial tumor formation, especially given that we and others found p53 mutations in tumors that always showed close resemblance to Group 3 MB[39]. This brain tumor type transition (Supplementary Fig. 7l) might not be that surprising as Glt1 show abundant expression in both forebrain and hindbrain cells and that depletion of such an important suppressor gene such as ARF might render more cell types sensitive to tumor formation. The balance of MYC and ARF is well coordinated and their levels regulate neuronal and glial fate choices in a developmental stage-dependent manner. Although MYC or MYCN must be involved in the tumor initiation process, we found that the MYC/MYCN pathway is suppressed in ArfARF-depleted HGG-like tumors. MYC (and MYCN) expression promotes neurogenesis while inactivation of MYC in NSCs isolated from embryonic brain attenuates self-renewal and induces gliogenesis[56]. An inactivation of p19ARF in these NSCs further resulted in attenuated astrocyte differentiation in the postnatal brain which might indicate a selection of gliomagenesis rather than an embryonal/

neuronal MB evolution when *Arf* is depleted and the MYC pathway activity is reduced in our models. Constitutive *Arf* depletion in mice is also known to cause spontaneous glioma rather than neuronal brain tumor formation[57]. Similarly, pediatric HGGs (pHGGs) in patients often present with *CDKN2A* loss and increased PDGF pathway signaling that was also found in the pHGGs induced in our mouse model[20,42,44].

We have shown through Confetti fate-tracing that tumors derived from the GMYC model establish from a dominant, monoclonal cell population – acquisition of secondary genetic drivers, in this case, MYC-driven suppression of *CDKN2A*, is putatively a tumorigenic event occurring in only a small subpopulation of cells, facilitating their survival and expansion. There are no targeted drugs that can specifically restore *CDKN2A* expression but the use of demethylating agents[58] that are well tolerated in patients[59,60], such as 5-Azacytidine, can restore ARF levels in MB cells and perhaps reactivate the failsafe program driven by this tumor suppressor gene. As 5-Azacytidine inhibits DNMT and methylation broadly, its actions cannot be used to specifically target ARF and the drug did unfortunately not provide any improved benefit in reducing tumor growth in vivo. However, our experiments show that ARF regulation is possible. In fact, ARF expression is perhaps also a necessity for maintenance of photoreceptor-positive pediatric brain tumors. A balance of MYC and ARF is present in Group 3 patients and it is likely this might create a photoreceptor-positive environment with activation of master regulators like OTX2, that is clearly lost or reduced in the majority of *Arf*-depleted HGG-like tumors. It was at least evident that tumors in our model system were either becoming neuronal, MYC positive with functional ARF and photoreceptor pathway activity, or glial, less MYC-dependent, *Arf*-depleted and photoreceptor pathway negative (Supplementary Fig. 7l).

HSP90 activation is known to facilitate DNMT1-driven DNA methylation[61] and was found to be significantly enriched in MYC-driven as compared to MYCN-driven mouse as well as human MBs. HSP90 inhibitors have previously been shown to be effective in SHH-dependent medulloblastoma models[62] and require a wild-type, functional p53 in mediating their efficacy in inducing apoptosis of tumor cells. Exosome proteomics further reveal an increased activity of HSP90 proteins particularly in exosome surfaces in Group 3 medulloblastoma[63] where we further identified significantly increased transcriptional elevation of HSP90 signaling. By inhibiting HSP90 using Onalespib, we saw ARF restoration as a specific response elicited in the MYC-driven GMYC cells. Similar results were seen when tumor-bearing mice were treated with Onalespib significantly extending the survival time of these mice when given as a monotherapy, and even providing a better outcome than the mice which underwent irradiation. When we knocked *Arf* out in GMYC tumors, Onalespib therapy failed and the global gene set enrichment of increased HSP90 pathway was abrogated, suggesting a link between HSP90 and ARF and that efficient HSP90 targeting in brain cancer cells required functional ARF. This link between successful HSP90 inhibitory treatment and requirement of functional ARF was further confirmed when testing Onalespib treatment on cell lines derived from the *Arf*-depleted GMYC model, where there was no observed response of HSF1-mediated heat-shock-response pathway activation following HSP90 inhibition. When conducting the same experiments in patient derived MB cell lines, we saw a mirrored response to the animal data, supporting the hypothesis that minor expression of *CDKN2A*, along with *MYC*-expression, is a requirement for a more sensitive response to HSP90 inhibition; at least in comparison to *MYCN*-driven tumors (Supplementary Fig. 7l). Patients with high MYC and low CDKN2A further showed a significant enrichment of HSF1-mediated heat shock response suggesting that they are most sensitive to this therapy. Given that Onalespib is a known radiosensitizer[64] and that it penetrates the BBB[65] it would be tempting to evaluate its tumor suppressive effect along with standard irradiation in MYC-driven Group 3 patients where efficient treatment options are rare.

## Methods

### Animals

All experiments were performed and complies with national guidelines and regulations, with the approvals (C105/16, 5.8.18-16350/2017 and 5.8.18-18303/2021) from the animal care and use committee at Uppsala University, Uppsala, Sweden.

Glt1-tTA mice were generated as previously described[17]. Glt1-tTA mice were crossed with TRE-MYC mice[30] (JAX stock #019376) where *MYC* is regulated by a tetracycline operator (tetO), resulting in a Glt1-tTA/TRE-MYC strain (GMYC). Tumors derived from this model arise from overexpression of the *MYC* gene localized to the brain via hindbrain specific Glt1-expressing cells. Tumor presentation in the GMYC model and allograft experiments was defined as a phenotypic change of stupor observed in the mouse and a visible protrusion on the head. For survival analysis and treatment studies in vivo, the maximal tumor burden permitted was not exceeded. GMYC mice were crossed with pTRE-H2BGFP mice[66] (JAX stock #005104), resulting in a GMYC/TetGFP strain where tumor cells express *GFP*. GMYC mice were crossed with (tetO-cre) LC1 mice[31] (JAX stock #006234). The TRE-CRE-LC1 construct included the *Luc1* gene, causing tumor cells to emit a luminescent signal following uptake and catabolism of D-luciferin. GMYC mice were crossed with *Arf*[FL/+] mice[41] (JAX stock #023323) where the *Arf* gene (exon-1β) can be specifically floxed out but where the linked *Ink4a* (*Cdkn2a*) gene is left intact. The resulting GMYC/ARF strain would have partial or complete knockout of the *CDKN2A/p19ARF* gene. In the experiments and for the comparisons presented, all models used have been backcrossed (at least 4–5 generations) into the FVB/N mouse strain. The genetic background was FVB/N for all transgenic animal crosses used.

### Transplantation studies and in vivo imaging

For experiments in where human PDXs or GTML orthotopic transplants were compared or grafted, Athymic Nude mice (obtained from Charles River) were used. Adult animals were used for most engraftment, except for studies presented in Fig. 2 where newborn (FVB/N) animals were also used. Immunocompetent postnatal (P0-P7) or adult FVB/N or Nude mice were anesthetized and cerebellar allografts generated by injection of 200,000 GMYC tumor cells into the cerebellum. Mice were monitored until phenotypic presentation of a brain tumor, at which point they were euthanized. At killing, whole brains were excised and biopsies taken for experimental procedures. For in vivo tumor imaging (IVIS) experiments involving GMYC mice, mice were injected with 75 mg/kg of D-luciferin (Perkin Elmer, 122796) and imaged under the highly-sensitive, cooled CCD camera, Nightowl II LB 983 In Vivo Imaging System (IVIS) (Berthold Technologies). Mice were determined to be saturated with D-luciferin and luminescent signal was at its maxima 5–15 min post-injection. Mice were imaged for a period of 45 s to 5 min depending on initial recorded signal intensity and emitted luminescent light was recorded and digitalized onto a brightfield capture of the mice. Regions of interest (ROI) defined the areas of signal and quantitative values of photons/cm²sec were recorded using IndiGo software (Berthold Technologies). IVIS measurement allowed for pre-phenotypic monitoring of tumor growth, assessment of tumor burden at phenotypic presentation, and to follow the effect of dox-mediated *MYC* suppression and subsequent tumor regression.

### Doxycycline treatment

For long-term in vivo doxycycline (dox) studies, dox-supplemented (625 mg/kg doxycycline) animal chow (Envigo, TD.01306) was given to tumor-presenting mice without other food sources. After 30 days of dox food, mice were returned to a normal diet and monitored for the rest of their lifetime to observe for any relapse. IP injections of 15 mg/kg dox (Sigma-Aldrich, D9891) were administered for acute (6 hour) treatments. For tumor initiation studies, two different experimental

cohorts were created. The first cohort of mice received dox chow from P0 to P30 and were then given normal chow for the rest of their lives and monitored for any tumor growth. For the second cohort, the breeding pair received dox food during the gestation period from E0 to E21/P0; after birth, pups received normal food for life and monitored for any tumor growth. For in vitro dox studies, cells were cultured in dox-supplemented media for up to two weeks and viability assessed via Alamar Blue assay during this period (Sigma-Aldrich, R7017). Each treatment was run in triplicate. Cells were continuously supplied with fresh media and dox. Viability readings were normalized against the DMSO control and untreated cells are those that received no treatment.

### Lineage tracing experiments

To lineage trace the GMYC tumor cells and determine clonality of tumors, GMYC mice were crossed with R26R-Confetti mice[36,67] (JAX stock #013731) to generate GMYC/Confetti. In the resultant animal strain, Tre-MYC-driven Glt1+ cells expressed 1 of 4 possible fluorescent proteins, conferring to the possible recombination events of the Confetti construct. It was not required to induce recombination due to inclusion of the intrinsic Tre-Cre within the model, thus the first recombination event during normal brain development and *MYC* activation would give rise to expression of one of these fluorescent genes. Brains of tumor-bearing mice were fixed by perfusion and frozen in O.C.T mounting compound. Tumor latency remained the same between the GMYC/Confetti mice and the GMYC mice (average age of 90–150 days at euthanasia). Brain tissue was sectioned at 12 μm thickness and counterstained with nuclear marker To-Pro-3 (Invitrogen, Cat #T3605). Immunofluorescent images were acquired using confocal microscope Zeiss LSM710-NLO equipped with a spectral detector. For spectral unmixing of the Confetti colors, the following lasers were used: 405 nm for CFP, 488 nm for GFP and YFP, 561 nm for RFP, 633 nm for To-Pro-3.

### In vivo demethylation and treatment of animals

For in vivo demethylation, HSP90 inhibition, and irradiation experiments, adult immunocompetent mice were injected intracranially with 100,000 GMYC/TetGFP/Luc cells and monitored via IVIS imaging until moderate tumor expansion. Prior to phenotypic symptoms, mice began treatment with either 5-Azacytidine (demethylation), Onalespib (HSP90 inhibition), or radiation. For demethylation studies, mice were administered daily I.P. injections of 0.2 mg/kg 5-Azacytidine (Abcam, ab142744). For HSP90 inhibition studies, mice were administered daily I.P. injections of 20 mg/kg Onalespib (Selleckchem, AT13387) prepared in 17.5% 2-hydroxypropyl-β-cyclodextrin (Sigma-Aldrich, H107) for 4 days after 4 days of recovery post-injection. For focal cranial irradiation, we used a dose of 2 Gy per day for 5 days (dose rate 1.0 Gy/min) with a high-resolution small animal radiation research platform (SARRP, Xstrahl Inc.). The radiation was performed under inhalational isoflurane anesthesia at the Preclinical Cancer Treatment Center, SciLifeLab, Uppsala University. In all instances, mice were imaged under IVIS during the treatment period until killed, in order to follow any tumor regression or expansion, and survival times were recorded.

### Immunostaining

Whole brain tissue was fixed in 4% formalin (VWR, 11699455) and sagitally halved prior to embedding in paraffin. Tissue sections were cut at 5 μm and mounted onto Superfrost Plus glass slides. Paraffin sections were deparaffinised, hydrated in graded ethanol solutions, and underwent heat-induced antigen retrieval. Sections were blocked and then incubated with secondary antibody, and visualized with DAB substrate according to manufacturer´s protocol for Vectastain ABC Elite kits (Vector Labs, SK-4105). Sections were counterstained in haematoxylin then mounted. To investigate leptomeningeal dissemination; at killing, the spinal cords were processed alongside the brain tissue and histologically analyzed for any tumor metastasis. Here, any clusters of >10 tumor cells found in the spinal cord (after checking 5 coronal spinal cord sections per tumor) was considered a metastatic spread. For each Immunohistochemical stain, >5 biological replicates were used to ensure reliability and accurate representation. Multiple histographs were taken from each replicate and a representative image was chosen for each figure.

### Senescence staining

Tumor-bearing mice were treated with dox and killed at predetermined time points. Brains were removed and stored in 30% sucrose solution overnight, prior to freezing in O.C.T mounting compound. Sections were cut at 10 μm thickness and immediately fixed in 4 °C formalin for 10 minutes. Following this, the slides were washed thrice in PBS and once in distilled water. Next, slides were incubated in X-gal working solution for 24 h at 37 °C. The following day, slides were rinsed twice in PBS and once in distilled water. Sections were mounted in aqueous mounting media and representative micrographs were taken demonstrating areas of tumor senescence.

### Western blot analysis

Tumor samples and/or cultured cells were dissociated mechanically prior to cell lysis. Complete cell lysis was carried out using the Diagenode Bioruptor Pico using a procotol of 30 s ON, 30 s OFF for 3 minutes total. Tumor lysate was loaded and resolved on 4-12% BIS-TRIS Nu-Page gels (Thermo Fisher), dry transferred onto Nitrocellulose membranes (Thermo Fisher), blocked in 5% BSA/TBS-t, and then probed for target proteins overnight at 4 °C. Protein-bound membranes were washed thrice in TBS-t and then incubated for 1 hour at room temperature with HRP-conjugated secondary antibody. Membranes were visualized under chemiluminescence. Biological replicates ensured reliability and accurate representation. For any experiments where multiple blots were used to represent one experiment, a BCA assay was conducted in order to accurately load equal amounts of protein lysate and all blots were conducted at the same time. Furthermore, for each blot, probing of a housekeeping protein was conducted to ensure equal load.

### Antibodies

Antibodies for immunostaining and western blotting were used as per the manufacturer's recommendations. The following antibodies were used within this study: β-tubulin (*Cell Signalling Technology*, Cat # 2146 (1:1000)), Cleaved Caspase-3 [Asp175] (*Abcam*, Cat # ab49899 (WB 1:500, IHC 1:1500)), CDKN2A/p19ARF (*Abcam*, Cat # ab80 (1:1000)), cMYC [Y69] (*Abcam*, Cat # ab32072 (WB 1:1000, IHC 1:500)), Cyclophilin B (*Cell Signalling Technology*, Cat # 43603 (1:1000)), GFAP [GA5] (*Millipore*, Cat # MAB3402 (1:1000)), GFP (*Abcam*, Cat # ab13970 (1:1000)), HSF1 (*Abcam*, Cat # ab61382 (1:1000)), HSP70 (*Abcam*, Cat # ab2787 (1:1000)), HSP90 [EPR16621] (*Abcam*, Cat # ab203085 (1:1000)), Ki67 [SP6] (*Abcam*, Cat # ab16667 (1:2000)), n-MYC [NCM II 100] (*Abcam*, Cat # ab16898 (1:250)), NPR3 (*Abcam*, Cat # ab97389 (1:250)), Olig-2 (*Millipore*, Cat # AB9610 (1:1000)), OTX2 (*R&D Systems*, Cat # AF1979 (1:500)), p21 (*Abcam*, Cat # ab109199 (1:1000)), Synaptophysin [SY38] (*Millipore*, Cat # MAB5258 (1:500)), TUJ1 (*Covance*, Cat # MMS-435P (1:500)), β-actin (Santa Cruz Biotechnology, Cat # sc-47778 (1:1000)), CDKN2A/p16INK4A (Abcam, Cat # ab211542 (1:1000)), anti-mouse IgG secondary HRP (VWR, Cat # NXA931 (1:1000)), anti-rabbit IgG secondary HRP (GE Healthcare, Cat # NA934 (1:1000)), and Lamin B1 (Abcam, Cat # ab16048 (1:1000)).

### Establishing tumor colonies in vitro

At killing, biopsies of tumor tissue were taken and mechanically dissociated into a single-cell homogenate. Cells were grown as immature neurospheres in Neurobasal media (Thermo Fisher, 10888022) containing 2% B-27 −A (Thermo Fisher, 125870-01), 1% PEST (Sigma-

Aldrich, P0781), 1% L-glutamine (Sigma-Aldrich, G7513), 20 ng/ml EGF (PeproTech, AF-100-15), 20 ng/ml bFGF (PeproTech, AF-100-18B), and maintained at 37 °C in an incubation chamber with an atmosphere of 5% CO$_2$. Cells were passaged bi-weekly. Cell lines were considered established if they retained high levels of MYC and remained responsive to dox-treatment, and these lines were expanded for use in subsequent in vitro experiments.

## Human cell lines
Patient derived human medulloblastoma lines D425 (Sigma-Aldrich, SCC290) as well as D283 and DAOY (ATCC, HTB-185 and HTB-186) were cultured in DMEM media supplemented with 10% FBS and 1% PEST (Sigma-Aldrich, P0781). Cells were cultured as monolayers on ultra-low attachment flasks and maintained at 37 °C in an incubation chamber with an atmosphere of 5% CO$_2$. These lines were chosen for subsequent cell proliferation and dose response assays and as human cell counterparts to our animal model cell lines due to their similarity – either in expressing MYC/MYCN or ARF status – transcriptional data regarding gene expression can be found within this paper and in our previously published work. Cell lines used were contamination-free when obtained from repositories or vendors but tested regularly upon culturing and passaging using a Mycoplasma detection kit, MycoAlert (Lonza). Only confirmed mycoplasma-free lines were used in research.

## DNA isolation
Primary tumor and matching spleen controls were excised and flash frozen on dry ice. DNA extraction was carried out using Qiagen's *DNeasy Isolation Kit* as per the manufacturer's instructions and stored for use in subsequent sequencing experiments.

## RNA isolation, cDNA preparation, and Trp53 sequencing
At killing, tumor biopsies were flash frozen in dry ice. Frozen biopsies were stored at −80 °C until RNA extraction. RNA extraction was carried out using Qiagen's RNeasy Isolation Kit as per the manufacturer's instructions. Purified RNA was submitted to SciLifeLab, Uppsala Genome Center, for RNA sequencing.

## Demethylation and treatment studies of cells
For in vitro 5-Azacytidine demethylation studies and subsequent protein analysis, cells were cultured in the presence of 0.01, 0.1, 1, 10 μmol/L 5-Azacytidine (Abcam, ab142744). Dose response curves were generated to determine EC50 concentrations used for subsequent viability assays, and these assays were conducted as previously described. Response [%] expresses the extent of cell death. Each treatment was run in triplicate. Cell lysates were prepared as previously described. For in vitro HSP90 inhibition studies, dose response curves were generated to determine EC50 concentrations used for subsequent viability assays. Combination Index (CI) values to determine synergism were calculated and scored according to the CompuSyn software. Cell lysates were prepared as previously described.

## Viral construct and cell transduction
c-MYC cDNA was cloned into pLenti6.3/TO/V5-DEST vector using Gateway cloning system to generate the lentiviral expression vector pLenti6.3-cMYC-V5. The MYC mutant construct (MYCV394D)[47] that is not able to bind MIZ1 was a kind gift from Prof. Dr. Martin Eilers. Infectious lentiviral particles were generated in HEK 293T cells as described previously[16]. After transduction of previously established MYCN-regulatable GTML2 cell lines[15] or MYC-regulatable GMYC cell lines, cells were selected with either blasticidin or puromycin to generate stable cell lines. Resultant lines were the original GTML tumor cells with transduced overexpression of *MYC*, termed GTML2/ + MYC; and GTML or GMYC parent lines with transduced overexpression of MYCV394D, termed GTML2/ + MYCVD or GMYC/ + MYCVD.

Established cell lines could now be maintained in Neurobasal media, as described previously.

## Mouse DNA methylation array and raw data processing
The methylation in DNA from GTML ($n = 5$) and GMYC ($n = 5$) tumor samples was profiled using the MM285 Infinium Mouse Methylation BeadChip, a platform designed to interrogate the methylation status of ~300,000 CpGs throughout the mouse genome[68]. Following raw data generation, all IDAT files were processed in R (v4.1.2) using the package SeSAMe (version 1.14.2)[69] and annotated with the MM285 Infinium Mouse Methylation Manifest 12v1-0 manifest. Using the manifest, probes known to be poor quality (e.g., cross-hybridizing, SNP-enriched) were masked and remaining data values normalized using normal-exponential out-of-band (noob) method[70]. Following normalization, probes with a detection p value >0.05 were removed and remaining probe intensity values converted into beta values for downstream analysis. All remaining probes were then annotated with genetic information using the mm10/GRCm10 mouse genome assembly where probes around the *Cdkn2a* genomic locus were further analyzed.

## RNA-sequencing data processing
Total RNA from GMYC ($n = 6$) tumor samples was sequenced through the Ion Proton™ Sequencer at the Uppsala Genome Center, SciLifeLab, Uppsala University. Raw RNA-seq reads GMYC, previously generated GTML tumors ($n = 7$)[32] and new GTML tumors and Arf knockout cell lines and controls were aligned to the mouse mm9 genome assembly using the STAR aligner (v2.7.2b)[71] followed by an additional application of Bowtie (v2.3.4.3)[72]. The STAR mapping utilized a two-pass approach, with the first alignment pass designed to discover splice junctions, which were then utilized for an improved alignment in the second pass. After mapping, the read counts for all genes or *Arf/Ink4a* transcripts were extracted using the feature-Counts function from the subread (v1.5.2) package[73] and utilizing mm9 annotations from GENCODE. The Ensemble gene ids were translated to official mouse gene symbols and orthologs were further mapped to their respective HGNC symbols using the biomaRt (v 2.42.1) package[74]. For downstream visualization and cross-species analyses, read counts were converted to measures of transcripts per million (TPM).

## Whole exome sequencing
24 GMYC and 12 GTML tumor biopsies, normal control sample pairs and GMYC1 and GTML2 tumor cell lines were whole exome sequenced by the National Genomics Infrastructure SNP&SEQ Technology Platform. Sequencing library preparation were performed using Twist Mouse Exome Panel Kit (Twist Bioscience). Clustering generation and paired-end sequencing were run for 100 cycles in one flowcell using the NovaSeq 6000 system (Illumina). Reads were aligned to the GRCm38.p6 reference genome build using Burrows-Wheeler Aligner version 0.7.17[75]. Bam files were converted using Samtools v. 1.14[76] and duplicated reads where marked using Picard v. 2.23.4 (https://broadinstitute.github.io/picard/). Variants in tumor-normal and tumor-only samples were called with VarScan v. 2.3.9[77] using somatic and mpileup2cns settings respectively. Somatic variants from tumor-only samples were obtained by excluding all variants reported in the normal samples. All samples were annotated using SnpEff v. 5.0C[78]. A detailed analysis of specific mutations and allele frequencies on the Trp53 transcript (ENSMUST00000108658.9) on Chr.11 was performed. The computations were partially enabled by resources provided by the Swedish National Infrastructure for Computing (SNIC) at UPPMAX, in part funded by the Swedish Research Council through grant agreement no. 2018-05973.

## Cdkn2a read coverage analysis

In order to estimate the read coverage of the exons corresponding to the *Arf/Ink4a* transcripts within GMYC and GTML tumor samples, we first extracted BEDGRAPH coverage files for the genomic region around *Cdkn2a* using the BEDTools (v2.27.1) package. Subsequently, we extracted the *Arf/Ink4a* transcript/exon locations from the UCSC browser and displayed the position-wise read coverage explicitly for those exons utilized by the *Arf* and/or *Ink4a* transcripts.

## External data sets

A processed microarray gene expression dataset comprising 763 MBs (GSE85217[33]) was downloaded from the Gene Expression Omnibus (GEO). Downstream cross-species classification analyses utilized only the 737 samples with the arguably most distinct transcriptional affiliation to a single subgroup[29]. Two processed data sets comprising mixtures of different brain tumors were downloaded either from GEO (GSE73038, microarray gene expression, $n = 182$[44,79], or the Children's Brain Tumor Network(CBTN www.cbtn.org, RNA-seq, $n = 996$[80]).The gene expression from the former was utilized as it was; from the latter only the TPM (transcripts per million) processed gene expression values were extracted. To reduce the complexity of these dataset for cross-species analyses, tumor types with fewer than 10 samples were removed prior to downstream analyses. Finally, a batch-normalized data set comprising 1350 MB and 291 normal cerebellar gene expression profiles was obtained from GEO (GSE124814[29]), 24 cerebellar samples of which were removed because they were not annotated by age.

## Determining high MYC/MYCN expression across samples

A rough estimate of the percentages of samples with high gene expression of either *MYC* or *MYCN* per MB subgroup or in normal cerebellum was calculated from the GSE124814 dataset. Specifically, the expression of both genes was first z-transformed across all samples. Subsequently, any sample with a $z$-score $z > 1$ for *MYC* or *MYCN* was considered to exhibit a high expression of that gene, respectively.

## DNA-methylation data analyses

Raw IDAT files for a DNA methylation dataset comprising 763 MBs (GSE85212[33]) were downloaded from the Gene Expression Omnibus (GEO) and pre-processed to beta values using the minfi package (v1.24.0) and the IlluminaHumanMethylation450kmanifest (v0.4.0) package. Differential methylation analyses, including significance testing, of *CDKN2A* specific CpG probes between MB samples were performed using the dmpFinder function from the minfi package.

## Identification of MB samples with putative *MYC* or *MYCN* amplifications

The identification of MB Group 3 and Group 4 samples with putative amplifications of *MYC* or *MYCN* was conducted as follows. A Methylation Set was derived from the raw GSE85212 IDAT files using the minfi package and annotations about different *MYC* or *MYCN* transcripts were extracted from the TxDb.Hsapiens.UCSC.hg19.knownGene (v 3.2.2) package. Subsequently, copy number intensities (CNIs) for each transcript were estimated using the conumee (v1.12.0) package by comparing the methylation of the respective sample to the WNT subgroup, which was employed as reference. CNIs were then averaged across all transcripts for each *MYC* and *MYCN*, respectively, and samples with an average CNI > = 0.25 were considered to exhibit a putative copy number amplification.

## Differential gene expression analyses

Differential gene expression analyses for both microarray expression and RNA-seq data were performed using the limma (v3.42.2) package[81]. For RNA-seq data, the read counts were first filtered to exclude non-expressed genes, such that only genes were included for which at least three samples had a CPM (Counts Per Million) value above 1, i.e. genes for which sum(cpm>1)>=3. Secondly, read counts were normalized with respect to the trimmed mean of M-values (TMM[82]) via the calcNormFactors function from the edgeR (v3.28.1) package[83], and then finally further processed using the voom function from the limma package. If otherwise not indicated, Box-plots for illustrating differential gene expression between groups of samples were generated in R using standard settings, such that the center line represents the median expression within the group, the box limits correspond to versions of the 1st and 3rd quartile, respectively, whiskers indicate the most extreme data points that are at most 1.5 times the interquartile region (IQR) above or below the box respectively, and points outside the whiskers are considered outliers.

## Cross-species analyses

Cross-species analyses were performed as previously described[16], with mouse tumor transcriptomes being mapped to human tumor transcriptomes in terms of a metagene projection. For cross-species classifications against the GSE85217 dataset, PCA plots and hierarchical clustering (HC) of metagene expression values were performed, using Euclidian distances and average linkage for the HC. Comparisons against the CBTN and GSE73038 datasets were instead concluded with a t-SNE clustering based on the metagene expression values.

## Drug selection for GMYC and GTML treatments

In order to identify drugs that might specifically elicit a response in GMYC rather than GTML tumors, we explored existing drug target gene sets in the CGP (Chemical and genetic perturbation) data base queried by our previous GSEA analysis between GMYC and GTML. We aimed for gene sets representing genes downregulated by drug treatment according to published studies, but the genes of which were upregulated in GMYC tumors as compared to GTML. Specifically, as these gene sets are enriched in GMYC and have been reported to be downregulated upon exposure to the drug, GMYC tumors might be particularly sensitive to the corresponding drug treatment. As a starting point, we selected the 10 top-significant drugs gene sets enriched in GMYC as compared to GTML. In order to further narrow down this list to drugs that might even be clinically relevant for human MBs, we then evaluated these 10 drug gene sets further by comparing them via a preranked GSEA between Group 3γ and Group 3 α + Group 4 tumors. The most promising drugs were then selected as those that were enriched with a normalized enrichment score NES > 4 in both GMYC as compared to GTML and Group 3γ as compared to Group 3α + Group 4 tumors.

## Patient survival analyses

Survival analyses between groups of human MB samples were conducted in R using the survival package. High and low groups were determined either using a single gene expression-derived cut-off based on the median or mean value, or alternatively using the 25% of samples with highest and 25% of samples with lowest expression, respectively. Statistical significance in survival between high and low groups was computed via the log-rank (Mantel-Haenszel) test as implemented in the included survdiff function. To avoid irrelevant survival effects caused by secondary tumors that might arise in long-term survivors, patients with survival times longer than 10 years were censored at exactly 10 years.

## Gene Set Enrichment Analyses

Gene Set Enrichment Analyses (GSEAs) were performed as previously described[16].

## Animal survival statistics

All animal experiments were conducted once and compared by using litter mates. Animal survival was graphically shown as a Kaplan–Meier curve, made and assessed using GraphPad Prism 7 software.

## Immunohistochemistry and immunofluorescence quantification

All micrographs shown are representative images of the respective mouse strain tumors. Where appropriate for statistical analysis and immunohistochemical quantification, at least three representative micrographs from at least three individual tumors were analyzed in Fiji ImageJ. Positivity of staining was expressed as a percentage of total cell population.

## Quantification of metastatic dissemination

Quantification of metastatic dissemination was assessed via histological examination of H&E-stained transverse sections of spinal cord. Each spinal cord was divided into several smaller, transverse regions prior to being processed, and multiple sections were taken per spinal cord. Metastatic dissemination was considered either positive or negative – where positivity was determined as a clear aggregate of 10 or more tumor cells. Numbers of analyzed animals are shown in the respective figures.

## Reporting of *p* values, sample size, and statistical significance

Significant *p* values are reported for appropriate figures either in the figure legend or at the respective figure panel. Sample size (*n*) can be found in the figure legend or in the respective figure panel. Error bars represent the standard error of mean.

## Randomization and inclusion/exclusion criteria

Animals included in the allograft studies had no inclusion/exclusion criteria prior to being stratified into treatment or vehicle groups. After killing, presence of physical tumor was confirmed by pathological examination and H&E staining in brain or spinal cord. Animals that were found dead without obvious signs of tumor (primary tumor penetrance studies) were marked as censored. No other data was excluded from analysis and reporting in this study. No blinding was carried out at any stage of the study.

## Reporting summary

Further information on research design is available in the Nature Portfolio Reporting Summary linked to this article.

## Data availability

Raw and processed RNA-seq data for the GMYC and GTML tumor samples have been deposited in the Gene Expression Omnibus (GEO) database and are available via the accession number GSE139240. Previously published data used in this study are available from GEO using accession numbers GSE85217 and GSE85212[33], GSE73038[44,79], GSE124814[29], GSE162080[31] and from Children's Brain Tumor Network[80] (https://cbtn.org/). All data supporting the findings of this study are available within the article and its supplementary information files. Source data are provided with this paper.

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

## Acknowledgements

This work was generously sponsored to FJS by the Swedish Childhood Cancer Foundation, the Swedish Cancer Society, the Swedish Research Council, the Swedish Brain Foundation, the Sjöberg Foundation and the Ragnar Söderberg's Foundation, to MK by the Swedish Cancer Society and Swedish Research Council and to SCC/RMH by Cancer Research UK and the Medical Research Council. We thank Dr. William Weiss (UCSF, San Francisco, USA) for providing us with the human MYC construct for cloning of the lentiviral MYC plasmid and Dr. Martin Eilers and Dr. Steffi Herold (University of Würzburg, Germany) for providing us with lentiviral MYC-MIZ mutant constructs. We further acknowledge technical support and usage of the BioVis facility, the National Genomics Infrastructure (NGI)/Uppsala Genome Center and UPPMAX for helping with massive parallel sequencing, and computational infrastructure. We acknowledge the Preclinical Cancer Treatment Center, SciLifeLab, Uppsala University and Dr. Tobias Bergström for technical assistance and support with radiation therapy experiments in vivo. Work performed at the National Genomics Infrastructure/Uppsala Genome Center has been funded by RFI/VR and the Science for Life Laboratory, Sweden. This research was conducted using data made available by the Children's Brain Tumor Network (CBTN). The computations and data handling were enabled by resources provided by the Swedish National Infrastructure for Computing (SNIC) at UPPMAX partially funded by the Swedish Research Council. Parts of the study were performed at the Live Cell Imaging facility, Karolinska Institutet, Sweden, supported by grants from the Knut and Alice Wallenberg Foundation, the Swedish Research Council, the Centre for Innovative Medicine, and the Jonasson center at the Royal Institute of Technology, Sweden. We finally thank Raquel Faguás Pérez and Jordi Peris Bravo for technical assistance.

## Author contributions

Conceptualization, F.J.S., H.W., and O.J.M.; Methodology, F.J.S. H.W., O.J.M., M.K., S.R., D.T., R.M.H., and S.C.C.; Investigation, O.J.M., H.W., S.H., G.R., A.B., L.B., A.D.V, A.S., K.H, K.A.,M.Z., K.O.H., S.R., and D.T.; Resources, F.J.S., R.M.H., and S.C.C.; Writing, O.J.M., F.J.S., H.W., and S.H.; Funding acquisition, F.J.S.; Supervision, F.J.S. and H.W.

## Funding

## Competing interests

The authors declare no competing interests.
