## [Peer Review File · Nature Communications]

ARF suppression by MYC but not MYCN confers increased malignancy of aggressive pediatric brain tumorsREVIEWER COMMENTS

Reviewer #1 (Remarks to the Author):

The manuscript presents interesting data using animal models of Group3 medulloblastoma, showing that MYC and MYCN activate tumorigenic programs that are similar but non- equivalent, and that the differences have important implications for targeted therapy. The quality of the data is high and the figures are excellent. The writing suffers from a lack of clarity. While few additional experiments are recommended, attention is needed to the way the results are presented and discussed. The introduction results and conclusions could be streamlined to present a more intentional story. The results section should be put into past tense, since it describes what actually happened. Specific areas where clarification is needed are detailed below. With revision to make the language more and concise, and attention to the specific issues below, the work will be an excellent contribution to the literature.

The absence of GFAP and OLIG2 staining is a significant finding that has important implications for the heterogeneity or absence of heterogeneity within the tumors. In patient samples, glial cells are often interspersed with tumor cells, and in animal models of SHH medulloblastoma, glial cells have been shown to derive from the tumor lineage. The data in Fig show absence of GFAP+ cells in the region of tumor shown. However, the OLIG2 staining in Fig does not look completely negative. Rather, there are a few cells with faintly+ nuclei. With IF staining, where the nuclear counterstain is a different channel, the nuclear OLIG2 might be more apparent. The authors should examine OLIG2 staining by IF, and determine if there are rare OLIG2+ cells interspersed in the tumors, and either way, the implications regarding cellular heterogeneity and similarity to human tumors should be discussed.

The method of deleting Cdkn2a should be more explicitly stated in the results section and additional validation may be needed to determine if heterozygosity effectively reduced expression. The authors should break up the run-on sentence "Having identified the differential expression of Arf in both models, we investigated whether modification of the Cdkn2a gene, encoding p19ARF, is involved in tumor formation in the GMYC model by generating partial and complete Cdkn2a (Arf gene-specific) knockout strains of our GMYC and GTML models." Breaking up this sentence into at least two sentences will improve clarity. Additionally, they should detail the genotypes used for partial and complete knockout of Cdkn2a. Whether heterozygosity generates partial knockdown is likely gene specific, and may be different in the tumor models. Western blot or qPCR should be used to show if Cdkn2a transcripts are different in GMYC vs GMYC/Arf HET and GTML vs GTML/Arf HET tumors. Without confirmation that Cdkn2a transcripts are lower in the Arf HETS, the ArfHET survival curves are difficult to interpret.

The data presented in 6e is not well-described in the text, which currently states "Treatment over 12 hours generated a response-gradient in the GMYC1 cells". A more direct phrasing, is recommended, such as "5-azacytidine treatment resulted in a dose dependent decrease in p19ARF expression in the GMYC cells but not in the GTML2 cells". Stating the result directly will improve clarity.

The conclusions of the studies with 5-azacytidine should be more clearly stated. The studies seem to show that methylation suppresses p19ARF expression in GMYC cells, and also that treating with 0.1 uM 5-azacytidine, which is sufficient to up-regulate p19ARF, does not impair cell viability in vitro. If these results are correctly stated, do the authors agree that these data argue that p19ARF suppression is not required for viability? If so, these implications of the data should be clearly stated. If the authors do not agree, the text should make clear their rationale for a different interpretation.

The different outcomes with 5-azacytidine and Onalespib are striking and need to be better discussed. Both 5-azacytidine (direct inhibition of DNA methylation) and Onalespib (HSP90 disruption) up-regulate p19ARF in GMYC, but only Onalespib slows tumor growth. Based on these data, it would seem that the mechanism for the anti-tumor effects of Onalespib is more complex than simply the re-expression of p19ARF. Put another way, the data show that up-regulating p19ARF is necessary (as shown by the lack of benefit of Onalespib in ARF-deleted tumors) but not sufficient for tumor inhibition. The observations could be explained if HSP90 inhibition activates the p53 pathway at multiple points, in which p19ARF is one point, but that other points (not also activated by 5-azacytidine) are also critical. If the authors agree with this interpretation, then making it clear will

prevent misinterpretation and confusion. If the authors do not agree with this interpretation, they should make an alternative interpretation that is clearly stated.

Abstract: "inexplicable" is not a good word choice. With more information, the phenomenon will not be inexplicable. Suggest either "paradoxical" or "seemingly inexplicable"

Intro line 39: suggest changing "MYC gene amplifications are often mutually exclusive..." to "MYC family gene amplifications are often mutually exclusive" or " MYC and MYCN gene amplifications are often mutually exclusive."

Intro lines 67-69. The grammar is awkward in the sentence "Benanti et al. have previously reported that down-regulation and methylation of ARF is propagating immortalization of human cells that is driven by MYC overexpression.", , which makes the meaning open to interpretation. The sentence should be re-phrased in an active voice, replacing "is propagating" with a direct indicative that has a clear subject.

Intro lines 70-2 Awkward phrasing should be simplified: "DNA methyltransferases (DNMTs) are 71 important mediators of maintaining normal genomic methylation" what are "mediators of maintaining"? better to say "DNMTs mediate" or "DNZMTs maintain"

Intro line 70-75. The logical steps from DNA methylation to DNMT inhibitors and HSP90 inhibitors to up-regulation of tumor suppressors should be stated more explicitly. This is a big idea that should be set apart in a separate paragraph.

Intro line 80, referring to MYC and MYCN as siblings is not correct. The idea seems like an extension of the gene family metaphor, but it would be better to be less figurative and more direct, by simply using "homologs"

Results 109, the authors need to clarify what is meant by a "phenotypic change of stupor?" Stupor is a state of altered mental status, slightly less severe than coma. "Change of stupor" is hard to parse.

Line 114 "curation" is used incorrectly to mean "cure"

Reviewer #2 (Remarks to the Author):

The authors study a mouse model of medulloblastoma driven by a MYC transgene and compare the results to a previous model in which they expressed a MYCN paralogue. They observe significant differences between tumors driven by either prologue and argue that these are due to differences in expression of the ARF tumor suppressor gene. They propose a model in which MYC can silence the expression of ARF whereas MYCN cannot not and suggest that pharmacological strategies that restore ARF expression may be useful for the therapy of medulloblastoma. In my view, this is an interesting manuscript. Below are a number of issues that the authors should address.

Comments

(1) In Figure 1c, the authors conclude that tumors depend on MYC continuously since they regress when MYC is switched off by addition of doxycycline after 30 days. This is important and interesting, but the authors do not show that the mice indeed had developed tumors before Dox was added. This control is essential.

(2) The legend of this figure (as well as others) contain strong conclusions, which by itself is unusual; also, some of them are not supported by the data. For example, MYC staining makes no statement about it being a key driving oncogene.

(3) It is unclear, which data support the statement in the text and the legend related to Figure 2 that tumors are embryonal. Also, the data shown in Figure 2 allow no statement about clonality.

(4) Figures 2c and 2d show highly abstract data; it would be good to know which genes drive the separation of tumor types in the principal component analysis in Figure 2d, for example. Which

functional categories of genes differ in expression between the tumors?

(5) Similarly, Figure 4a shows very few genes as different whereas Figure 4b shows very large gene groups as being different between MYC and MYCN-driven tumors. If Figure 4a shows gene sets, it is unclear what is plotted.

(6) Figure 4b shows that MYC-driven tumors are more active in driving expression of canonical MYC target genes and cell cycle genes than MYCN-driven tumors. This is interesting and surprising, since there are other tumors (like neuroblastoma) where it is the other way around. Since ARF has been suggested to interact with MYC proteins, it would be important to see whether loss of ARF enhances MYCN-driven transcription in GTML tumors. In other words, is gene expression in GTML/ARF-deficient tumors similar or identical to MYC tumors?

(7) In Figure 6, in particular in Figure 6c, chromatin occupancy of the INK4a locus needs to be shown for both MYC and MYCN.

(8) The best understood function of ARF is to activate p53; are the ARF-deficient tumors p53 wildtype?

(9) Figure 6d lacks a control by how much wtMYC reduces ARF expression in GTML2 cells to make the statement that MIZ1 is not involved. Also RNA data should be shown.

(10) I am not sure I understand how Hsp90 inhibitors act in this context. The authors make strong statements about this (Top of page 20), but do not show any data. Does Hsp90 inhibition affect MYC-dependent transcription profiles?

Reviewer #3 (Remarks to the Author):

This manuscript details an investigation into the molecular pathogenesis of MYC-driven medulloblastoma. The authors employed multiple mouse models to delineate molecular distinctions between MYC and MYCN-driven medulloblastoma, with a particular focus on the differential role played by ARF silencing in each. The authors also provided multiple correlations to relevant profiling data from human medulloblastomas. Finally, they demonstrated that HSP90 inhibition represents a potential therapeutic strategy for MYC-driven medulloblastoma and appears to operate through re-induction of ARF.

This manuscript represents a considerable body of work and introduces novel murine modeling reagents, which provide insights into disease biology and therapeutic strategy. My comments are below.

1) While it is somewhat disappointing that the authors have not definitively linked ARF repression in MYC-driven medulloblastoma to epigenetic silencing, the data they have are suggestive and they have apparently gone to considerable effort, unsuccessfully, to bolster it. Strengthening links between HSP90 efficacy and ARF induction would be helpful. In this regard, the authors should knockdown ARF in their MYC line treated with HSP90 inhibitor. They appear to have done a similar experiment using MYCN-driven lines, but for some reason didn't do this for MYC-driven lines.

2) The authors claim synergy between HSP90 inhibition and cisplatin in the treatment of their models. However, the presented data appear additive at best. True synergy should be more rigorously established with isobolographic analysis on an equivalent.

3) Minor: FIG.7-unless I'm mistaken, there is no cisplatin data presented. The authors should adjust the figure title accordingly.

4) Line 147: Craniopharyngioma is not a neuronal tumor. It is actually an epithelial neoplasm arising from the remnants of Rathke's pouch.

5) Line 164: The title of this section implies that the described work was performed in humans, not mice. This should be adjusted.

6) There are several minor grammatical issues, e.g. "tumor curation" instead of "tumor cure" and the inappropriate use of "which" instead of "that" on at least one occasion. Otherwise the manuscript reads quite well.

Point-By-Point response to REVIEWER COMMENTS on Mainwaring et al. 2022

Reviewer #1 (Remarks to the Author):

The manuscript presents interesting data using animal models of Group3 medulloblastoma, showing that MYC and MYCN activate tumorigenic programs that are similar but non- equivalent, and that the differences have important implications for targeted therapy. The quality of the data is high and the figures are excellent. The writing suffers from a lack of clarity. While few additional experiments are recommended, attention is needed to the way the results are presented and discussed. The introduction results and conclusions could be streamlined to present a more intentional story. The results section should be put into past tense, since it describes what actually happened. Specific areas where clarification is needed are detailed below. With revision to make the language more and concise, and attention to the specific issues below, the work will be an excellent contribution to the literature.

We thank the reviewer for the kind words regarding our manuscript and work. We have carefully combed through the results section and adjusted any instances of incorrect tense, as well as tidying up the language within the rest of the document. We hope that these changes have helped to clarify the data contained within.

The absence of GFAP and OLIG2 staining is a significant finding that has important implications for the heterogeneity or absence of heterogeneity within the tumors. In patient samples, glial cells are often interspersed with tumor cells, and in animal models of SHH medulloblastoma, glial cells have been shown to derive from the tumor lineage. The data in Fig show absence of GFAP+ cells in the region of tumor shown. However, the OLIG2 staining in Fig does not look completely negative. Rather, there are a few cells with faintly+ nuclei. With IF staining, where the nuclear counterstain is a different channel, the nuclear OLIG2 might be more apparent. The authors should examine OLIG2 staining by IF, and determine if there are rare OLIG2+ cells interspersed in the tumors, and either way, the implications regarding cellular heterogeneity and similarity to human tumors should be discussed.

Above are the OLIG2 IF stainings as requested. From these results, we stand by the statement that the majority of cells within GMYC tumors are OLIG2 negative. In the tumor bulk, very few OLIG2+ cells are seen, and the majority of positive cells can be found interspersed along the tumor border. These are potentially resident OLIG2+ cells which have been 'pulled' into the tumor during tumor expansion and growth. It is also possible, as suggested by the reviewer, that a rare population of OLIG2+ tumor cells reside within the tumor mass.

Historically, OLIG2 was rarely used as a marker for medulloblastoma, instead this marker was commonly restricted to identification of gliomas. However, the Pei lab, Children's National, DC in US have recently shown (Xu et al. *Clin.Can.Res.*, 2022) that OLIG2 positive cells can indeed be found in a subset of MB patients. Due to lack of space, we have not included the stainings shown above in the manuscript, but we added an additional sentence to address this when discussing these results (lines 136-138) to clarify how the staining pattern seen in GMYC tumors reflects the staining pattern seen in patient tumors.

The method of deleting *Cdkn2a* should be more explicitly stated in the results section and additional validation may be needed to determine if heterozygosity effectively reduced expression. The authors should break up the run-on sentence "Having identified the differential expression of *Arf* in both models, we investigated whether modification of the *Cdkn2a* gene, encoding p19ARF, is involved in tumor formation in the GMYC model by generating partial and complete *Cdkn2a* (*Arf* gene-specific) knockout strains of our GMYC and GTML models." Breaking up this sentence into at least two sentences will improve clarity. Additionally, they should detail the genotypes used for partial and complete knockout of *Cdkn2a*. Whether heterozygosity generates partial knockdown is likely gene specific, and may be different in the tumor models. Western blot or qPCR should be used to show if *Cdkn2a* transcripts are different in GMYC vs GMYC/*Arf* HET and GTML vs GTML/*Arf* HET tumors. Without confirmation that *Cdkn2a* transcripts are lower in the *Arf* HETS, the *Arf*HET survival curves are difficult to interpret.

We thank the reviewer for this suggestion and have modified this section in the results and clarified the crossing scheme. We have in better detail studied *Cdkn2a* levels (using Western blots) in a number of tumor biopsies from GMYC, GMYC/*Arf* HET and KO mouse tumors (see new Supp. Fig.5a). It is here clear from at least 3 biopsies in each group that ARF is highest in ARF +/+, partially suppressed in *Arf*+/- and not present at all in any of the *Arf*-/- biopsies. It is also clear from these data that *Arf* can be spontaneously lost also during the "normal" development of these tumors. This pattern of ARF loss follows a common theme in MYC-driven cancer evolution (Zindy et al. *Genes & Dev.*, 1998) and as we show in this manuscript (Supp. Table 1), p53 is often mutated during this process (40% of GMYC and GTML cases).

Finally, we studied DNA methylation of the *CDKN2A* locus using Illumina Infinium BeadChips and found that probes in the *CDKN2A* locus (around both *Arf* and *Ink4a*) are partially or fully methylated at various sites in almost half of GMYC and GTML samples but with no difference in methylation in GMYC as compared to GTML samples (See new Supp. Fig. 4c).

The data presented in 6e is not well-described in the text, which currently states "Treatment over 12 hours generated a response-gradient in the GMYC1 cells". A more direct phrasing, is recommended, such as "5-azacytidine treatment resulted in a dose dependent decrease in p19ARF expression in the GMYC cells but not in the GTML2 cells". Stating the result directly will improve clarity.

We thank the reviewer for this suggestion and have (on new line 411-12) modified the description to: “5-azacytidine treatment resulted in a dose dependent increase in p19ARF expression in GMYC cells but not in the GTML2 cells”.

The conclusions of the studies with 5-azacytidine should be more clearly stated. The studies seem to show that methylation suppresses p19ARF expression in GMYC cells, and also that treating with 0.1 uM 5-azacytidine, which is sufficient to up-regulate p19ARF, does not impair cell viability in vitro. If these results are correctly stated, do the authors agree that these data argue that p19ARF suppression is not required for viability? If so, these implications of the data should be clearly stated. If the authors do not agree, the text should make clear their rationale for a different interpretation.

We are sorry if the results part is not clear in Figure 6. 5-AZA treatment does significantly impair cell viability of GMYC cells in vitro (see Figure 6f-g) but not in vivo (see Figure 7h). GTML cell viability is however also suppressed by 5-AZA at a slightly higher concentration (see Figure 6f-g), suggesting that this blunt demethylation therapy is not specific for GMYC tumor treatment.

As stated in the previous comment, we now checked the methylation status in a number of GMYC and GTML tumors using Illumina methylation arrays. Methylation is indeed found in GMYC and GTML tumors at and around the CDKN2A loci on Chromosome 4 suggesting that methylation still could be one mechanism for ARF suppression in GMYC and GTML cells. Arf expression is still significantly lower in GMYC tumors in both human and mouse tumors, so while we toned down the epigenetic mechanism of Arf suppression in the paper we instead focused more on the mechanism behind the HSP90 inhibition and ARF suppression. See further next point below.

The different outcomes with 5-azacytidine and Onalespib are striking and need to be better discussed. Both 5-azacytidine (direct inhibition of DNA methylation) and Onalespib (HSP90 disruption) up-regulate p19ARF in GMYC, but only Onalespib slows tumor growth. Based on these data, it would seem that the mechanism for the anti-tumor effects of Onalespib is more complex than simply the re-expression of p19ARF. Put another way, the data show that up-regulating p19ARF is necessary (as shown by the lack of benefit of Onalespib in ARF-deleted tumors) but not sufficient for tumor inhibition. The observations could be explained if HSP90 inhibition activates the p53 pathway at multiple points, in which p19ARF is one point, but that other points (not also activated by 5-azacytidine) are also critical. If the authors agree with this interpretation, then making it clear will prevent misinterpretation and confusion. If the authors do not agree with this interpretation, they should make an alternative interpretation that is clearly stated.

We welcome this insightful comment from the reviewer and have now further investigated and compared the mechanisms of 5-AZA and Onalespib on GMYC tumor viability.

When GMYC tumors lose ARF by knock out, the MYC and NMYC gene sets but also photoreceptor activity are decreased, especially this is seen in the non-MB Arf knock-out tumors (See Supp. Figure 5 and our summary graph in Supp. Figure 7I).

Still, Onalespib treatment further induces a heat shock signature as we previously showed by a distinct gene set enrichment of “Regulation of HSF1-mediated heat shock response”. Interestingly, this gene set is enriched in CDKN2A low MYC high Group 3-alpha, 3-gamma and Group 4 patients as compared to CDKN2A high MYC low patients correlating heat shock activity to ARF suppression also in patients. We have presented this new data in Figure 5d and modified our conclusions in the paper.

Abstract: “inexplicable” is not a good word choice. With more information, the phenomenon will not be inexplicable. Suggest either “paradoxical” or “seemingly inexplicable”

We agree with the reviewer that this was not the correct word choice and have changed it accordingly.

Intro line 39: suggest changing “MYC gene amplifications are often mutually exclusive...” to “MYC family gene amplifications are often mutually exclusive” or “MYC and MYCN gene amplifications are often mutually exclusive.”

We have changed this sentence to prevent further confusion regarding the gene family.

Intro lines 67-69. The grammar is awkward in the sentence “Benanti et al. have previously reported that down-regulation and methylation of ARF is propagating immortalization of human cells that is driven by MYC overexpression.”, which makes the meaning open to interpretation. The sentence should be re-phrased in an active voice, replacing “is propagating” with a direct indicative that has a clear subject.

We agree with the reviewer that the written sentence was perhaps confusing and interpretation could be taken in multiple ways. Changes have been made to the sentence and we hope that the re-phrasing is clearer.

Intro lines 70-2 Awkward phrasing should be simplified: “DNA methyltransferases (DNMTs) are 71 important mediators of maintaining normal genomic methylation” what are “mediators of maintaining”? better to say “DNMTs mediate” or “DNZMTs maintain”

We agree with the reviewer and have removed this phrasing from the intro.

Intro line 70-75. The logical steps from DNA methylation to DNMT inhibitors and HSP90 inhibitors to up-regulation of tumor suppressors should be stated more explicitly. This is a big idea that should be set apart in a separate paragraph.

Thanks for this suggestion. We have toned down in the manuscript how HSP90 inhibitors are up-regulating TSG like ARF as we believe this is only one part of the action of these drugs that work more via a general heat shock response pathway with a specificity to MYC high CDKN2A low tumor types. We have therefore removed some of this phrasing in the introduction as well. We hope this makes sense.

Intro line 80, referring to MYC and MYCN as siblings is not correct. The idea seems like an extension of the gene family metaphor, but it would be better to be less figurative and more direct, by simply using “homologs”

We thank the reviewer for pointing out this error and have fixed the nomenclature accordingly.

Results 109, the authors need to clarify what is meant by a “phenotypic change of stupor?” Stupor is a state of altered mental status, slightly less severe than coma. “Change of stupor” is hard to parse.

We agree and now changed (see new lines 128-130) to “Tumor presentation was defined by a logarithmic increase in bioluminescent signal intensity, continuous weight loss observed and commonly with a visible protrusion on the head of the mouse.”

Line 114 “curation” is used incorrectly to mean “cure”

We agree that this was the wrong word choice and this error has been fixed.

Reviewer #2 (Remarks to the Author):

The authors study a mouse model of medulloblastoma driven by a MYC transgene and compare the results to a previous model in which they expressed a MYCN paralogue. They observe significant differences between tumors driven by either prologue and argue that these are due to differences in expression of the ARF tumor suppressor gene. They propose a model in which MYC can silence the expression of ARF whereas MYCN cannot not and suggest that pharmacological strategies that restore ARF expression may be useful for the therapy of medulloblastoma. In my view, this is an interesting manuscript. Below are a number of issues that the authors should address.

Comments

(1) In Figure 1c, the authors conclude that tumors depend on MYC continuously since they regress when MYC is switched off by addition of doxycycline after 30 days. This is important and interesting, but the authors do not show that the mice indeed had developed tumors before Dox was added. This control is essential.

To show that mice have developed tumors prior to dox treatment, we included Supplemental Figure 2g where we image a tumor-bearing mouse using an in vivo imaging system, and then follow tumor regression during the dox treatment. Most, but not all of our GMYC mice had intrinsic expression of Luciferase, meaning not all of our dox treated mice could be imaged in this manner – however, all mice were judged equally through phenotypic evaluation (weight loss, piloerection, social isolation, hunched posture, abnormal gait, protrusion on the head) to define the ‘end-point’ (either sacrifice or dox-treatment start) of the study.

We have followed more than 100 GMYC mice that developed tumors upon symptoms at this end point and have confirm by H&E that most of them had tumors (with just a few exceptions (3-4 out of 100 mice) that died of other causes). We also have video footage of the very same tumor-bearing mouse we followed prior to (Video_GMYC_mouse 6569_found sick_Nov18) and post-treatment (Video_GMYC_mouse 6569_3d green dox chow_Nov21) that we have not included in the supplemental data, but are happy to show the reviewer here. This is a typical/representative example of mice developing tumors in the GMYC model that has been “cured” after eating chow with high dox concentration (labelled green as compared to our normal chow without dox).

(2) The legend of this figure (as well as others) contain strong conclusions, which by itself is unusual; also, some of them are not supported by the data. For example, MYC staining makes no statement about it being a key driving oncogene.

Agree. This sentence has now been modified instead confirming that the TRE-MYC construct is functional when the human MYC protein is present. We also toned down our conclusions in other figure legends accordingly.

(3) It is unclear, which data support the statement in the text and the legend related to Figure 2 that tumors are embryonal. Also, the data shown in Figure 2 allow no statement about clonality.

We thank the reviewer for pointing out this discrepancy between the figure and the figure legend and we have consequently fixed this. This was a mistake that arose when we moved the clonality analysis from Figure 2 to Figure 3 just before submission.

(4) Figures 2c and 2d show highly abstract data; it would be good to know which genes drive the separation of tumor types in the principal component analysis in Figure 2d, for example. Which functional categories of genes differ in expression between the tumors?

The cross species analysis keeps a good separation of human MB subgroups while still being able to compare the predicted subtype of mouse tumors. It divides the model data into four meta genes, one per MB subtype, where each meta gene consists of a linear combination of all genes on the dataset. Plotting the resulting gene factors of signature genes of class-predicted medulloblastomas (Northcott et al. Acta Neuropath. 2012) as a heatmap (see below) indicate that the discriminating capability is retained in these separated genes.

Metagene model factors for MB signature genes (Northcott et al. Acta Neuropath. 2012)

Only conserved genes that are shared between mouse and human species are maintained in the dataset (in a similar way as we most recently published in Borgenvik et al. *Cancer Res.*, 2022). Typical functional gene categories for GMYC and GTML tumors have been further discussed in Figure 4b.

(5) Similarly, Figure 4a shows very few genes as different whereas Figure 4b shows very large gene groups as being different between MYC and MYCN-driven tumors. If Figure 4a shows gene sets, it is unclear what is plotted.

We can understand that the figure has an unclear interpretation given that there was an error in the figure legend regarding the source of the gene sets analyzed. This is now corrected and the figure used a targeted gene signature of either MYC or MYCN high genes found in Supp. Fig. 3d. It is further in all other aspects analogous to Fig. 2f.

(6) Figure 4b shows that MYC-driven tumors are more active in driving expression of canonical MYC target genes and cell cycle genes than MYCN-driven tumors. This is interesting and surprising, since there are other tumors (like neuroblastoma) where it is the other way around. Since ARF has been suggested to interact with MYC proteins, it would be important to see whether loss of ARF enhances MYCN-driven transcription in GTML tumors. In other words, is gene expression in GTML/ARF-deficient tumors similar or identical to GMYC tumors?

This is a valid point. No enrichment of typical MYC gene sets are found following ARF loss. MYC gene set activity is by contrast suppressed in ARF-depleted tumors (See new Supp. Figure 5h). This aspect is further discussed now in the paper. There is a link in brain tumor development and embryonal neural stem cell differentiation that MYC (and MYCN) promotes/favors neurogenesis/neuronal features and when MYC/MYCN are silenced it rather drifts towards the glial lineage. Constitutive Arf null mice are tumor prone and (apart from sarcomas and lymphomas) spontaneously develop gliomas but not any neuronal tumors (Kamijo et al. Cancer Research, 1999), which is now further discussed in the manuscript.

Agree, it is tempting to ask if GTML/ARF-deficient tumors are more similar to GMYC tumors but it seems like it is not only the ARF levels that are important here. It is more important to see if the tumor is an MB-like or an HGG-like (also described as non-MB in the manuscript) as gene expression in GTML/ARF-deficient tumors is sometimes similar to MB - but more commonly found as HGG-like (non-MB) tumors. See also our summary figure (Supp Fig. 7l).

(7) In Figure 6, in particular in Figure 6c, chromatin occupancy of the INK4a locus needs to be shown for both MYC and MYCN.

We performed an Illumina Infinium Mouse Methylation BeadChip array analysis of the CDKN2A locus for GTML and GMYC tumors. As seen in Supp. Fig. 4c, methylation of both Ink4a and Arf transcripts are commonly found in both GMYC as well as GTML tumors suggesting such silencing is likely involved in these tumors. See also additional Western Blots in Fig. 6c and an additional WB for Ink4a specific protein in Supp. Fig 6a.

(8) The best understood function of ARF is to activate p53; are the ARF-deficient tumors p53 wildtype?

Most ARF-deficient tumors were p53 wildtype. However, one (out of 4 biopsies sequenced) ARF-deficient tumor was p53 mutated. Still this is indicating that these are not exclusive events of gene loss/mutation and in some of these individuals there needed to be suppression of both genes. This is

in line with the report from e.g. Datta et al. *JBC*, 2004 that shows that ARF interacts with c-Myc independently of MDM2 or p53.

(9) Figure 6d lacks a control by how much wtMYC reduces ARF expression in GTML2 cells to make the statement that MIZ1 is not involved. Also RNA data should be shown.

This control figure of ARF protein levels is provided in a time-dependent manner in Supp Fig. 6c. Hope this is sufficient info provided for this.

(10) I am not sure I understand how Hsp90 inhibitors act in this context. The authors make strong statements about this (Top of page 20), but do not show any data. Does Hsp90 inhibition affect MYC-dependent transcription profiles?

Thanks for this comment. No, Hsp90 inhibition is not affecting the most commonly used hallmarks of MYC-dependent transcriptional profiles. However, upon Hsp90 inhibition of MYC-driven GMYC tumors, the gene set `REGULATION_OF_HSF1_MEDIATED_HEAT_SHOCK_RESPONSE` is upregulated along with e.g. p21 upregulation and BCL2-dependent apoptosis (see details of significantly enriched gene sets in Supp. Table 2). Viability data from in vitro and survival data from in vivo experiments are also seen in Fig. 7e, f, h, j and k.

Importantly, as we show in new Fig. 7d, CDKN2A low MYC high Gr. 3 patients have a significant enrichment in HSF1-mediated heat shock response as compared to CDKN2A high MYC low Gr. 3 patients. This correlation is not found if patients are similarly sorted after MYC or CDKN2A levels alone – e.g. MYC high versus MYC low patients has no significant correlation with heat shock response. When ARF is depleted the significance with `HEAT_SHOCK_RESPONSE` but also with photoreceptor pathway activity is lost and HSP90 inhibitors are not efficient anymore (Supp. Fig. 7I).

Reviewer #3 (Remarks to the Author):

This manuscript details an investigation into the molecular pathogenesis of MYC-driven medulloblastoma. The authors employed multiple mouse models to delineate molecular distinctions between MYC and MYCN-driven medulloblastoma, with a particular focus on the differential role played by ARF silencing in each. The authors also provided multiple correlations to relevant profiling data from human medulloblastomas. Finally, they demonstrated that HSP90 inhibition represents a potential therapeutic strategy for MYC-driven medulloblastoma and appears to operate through re-induction of ARF.

This manuscript represents a considerable body of work and introduces novel murine modeling reagents, which provide insights into disease biology and therapeutic strategy. My comments are below.

1) While it is somewhat disappointing that the authors have not definitively linked ARF repression in MYC-driven medulloblastoma to epigenetic silencing, the data they have are suggestive and they have apparently gone to considerable effort, unsuccessfully, to bolster it. Strengthening links between HSP90 efficacy and ARF induction would be helpful. In this regard, the authors should

knockdown ARF in their GMYC line treated with HSP90 inhibitor. They appear to have done a similar experiment using MYCN-driven lines, but for some reason didn't do this for MYC-driven lines.

We now performed methylation analysis of GMYC and GTML tumors using recently developed Illumina Infinium Mouse Methylation BeadChip arrays. We can here show that CDKN2A is commonly methylated in the mouse brain tumors. Still, the data does not show that ARF is specifically silenced in MYC tumors, as also MYCN-driven tumors show methylation in the locus. Therefore, we now toned down that methylation is critical for MYC-driven tumor development in our manuscript.

That said, CDKN2A and especially ARF is not as highly expressed in MYC-driven as compared to MYCN-driven MB patients and this is equally seen in our mouse models as we clearly show in this paper. We have therefore worked on strengthening the link between HSP90 efficacy and ARF further, and can show that MYC needs CDKN2A in order to generate Photoreceptor positive tumors and that this balance is important for a sustained heat shock response (see our summary in Supp. Fig. 7I). This is shown by the fact that CDKN2A low MYC high Gr. 3 patients have a significant enrichment in HSF1-mediated heat shock response as compared to CDKN2A high MYC low Gr. 3 patients (see new Figure 7d.). This correlation is not found if patients are similarly sorted after MYC or CDKN2A levels alone – e.g. MYC high versus MYC low patients has no significant correlation with heat shock response.

Perhaps the reviewer missed this data, but we did knock down ARF in the GMYC line and treated with HSP90 inhibitor and presented that it renders no difference in survival (in Supp Fig. 7e). This indicates that ARF is indeed needed for Onalespib therapy to work. Such ARF-depleted tumors also lose their photoreceptor pathway activity (Supp. Figure 5f). In addition, DAOY cells that are photoreceptor negative, also lack ARF expression and show resistance to Onalespib treatment. Hope all this information sounds informative and is convincing.

2) The authors claim synergy between HSP90 inhibition and cisplatin in the treatment of their models. However, the presented data appear additive at best. True synergy should be more rigorously established with isobolographic analysis on an equivalent.

We did not include any graphical isobolographic analysis in the paper and thank the reviewer for this suggestion. We went ahead and calculated synergism between multiple different combinations of Onalespib/Cisplatin concentrations and the combination index (CI) was calculated and scored according to the 'CompuSyn' software - from which it indicated synergism in the GMYC1 cells. Please see the isobolographic graph below – GMYC1 cells were treated with the EC50 concentration of Onalespib and a range of cisplatin concentrations and the combination treatment was evaluated to be synergistic for several of these concentration combinations treatments. This graph has been inserted into the manuscript in a new Supplemental Figure 7c.

3) Minor: FIG.7-unless I'm mistaken, there is no cisplatin data presented. The authors should adjust the figure title accordingly.

We thank the reviewer for pointing out this mistake and have made adjustments to the figure title.

4) Line 147: Craniopharyngioma is not a neuronal tumor. It is actually an epithelial neoplasm arising from the remnants of Rathke's pouch.

Thanks for this comment. We now renamed these group of tumors of various different entities to "other non-glial tumors". The idea with this comparison was just to show that these tumors present entities very far away from what we modelled in this paper.

5) Line 164: The title of this section implies that the described work was performed in humans, not mice. This should be adjusted.

We have changed the title of this section to better reflect that this data was generated from our mouse model.

6) There are several minor grammatical issues, e.g. "tumor curation" instead of "tumor cure" and the inappropriate use of "which" instead of "that" on at least one occasion. Otherwise the manuscript reads quite well.

We thank the reviewer for pointing out this incorrect usage of wording and have made efforts to fix these instances throughout the manuscript.

REVIEWERS' COMMENTS

Reviewer #1 (Remarks to the Author):

The authors have addressed my comments effectively and the manuscript is ready for publication.

Reviewer #2 (Remarks to the Author):

The authors have responded well to most of the comments. This is an interesting and important new mouse model for medulloblastoma.

What still puzzles me, why the authors insist that MYC-dependent repression is independent of MIZ1. This concerns several level of their data: I asked for ChIP data for MYC promoter occupancy (comment 7) and RNA data for ARF (comment 9) and the authors did not provide them, so how direct any of this is a bit open. ARF protein stability is highly regulated in itself, so this may well affect the results of these blots.

More importantly, Figure 6d shows exactly the reported phenotype for MYCV394D. In many cells, MYCVD is expressed at much higher levels than wtMYC, since it does not repress anti-apoptotic proteins (and hence high levels of MYCVD are much better tolerated than wtMYC), and levels of repressed genes are nevertheless still higher than in wtMYC cells. So relative to the amount of MYC protein, repression by MYCVD is clearly compromised- exactly in line with all biochemical data that show reduced, but not completely absent interaction of MYC VD to MIZ1. The fact that p19ARF levels go down with time in Supplementary Figure 6D is actually quite a concern for their experimental system (according to their model, ARF should be absent in GMYC1 cells from the get-go and definitely it should not go down when total levels of MYC go down, be it VD or wild-type). It is maybe also worth noticing that the other genes they see as being differentially expressed (cilium-related genes) are also MIZ1 target genes, as shown by Roussel and colleagues.

So based on the data, I think the authors need to remove the claim that this is MIZ1-independent phenomenon and - unless I misunderstand it - also Figure Supplementary 6D. Having done so, this will be a very interesting paper.

Reviewer #3 (Remarks to the Author):

I have no further issues with this manuscript.

Point-by-point response for NCOMMS-22-11592A; Additional response for Reviewer 2:

The authors have responded well to most of the comments. This is an interesting and important new mouse model for medulloblastoma. What still puzzles me, why the authors insist that MYC-dependent repression is independent of MIZ1. This concerns several level of their data: I asked for ChIP data for MYC promoter occupancy (comment 7) and RNA data for ARF (comment 9) and the authors did not provide them, so how direct any of this is a bit open. ARF protein stability is highly regulated in itself, so this may well affect the results of these blots.

We thank the reviewer for these comments. We only suggested that the repression on ARF (but nothing else) seemed independent on MIZ1. We agree with the reviewer and we did not have capacity to study global RNA effects or ARF in detail, or MYC promoter occupancy by ChIP in this study. We are happy to tone this independence of MIZ1 down throughout the manuscript, which we now have done in both abstract, introduction and in our results/discussions.

More importantly, Figure 6d shows exactly the reported phenotype for MYCV394D. In many cells, MYCVD is expressed at much higher levels than wtMYC, since it does not repress anti-apoptotic proteins (and hence high levels of MYCVD are much better tolerated than wtMYC), and levels of repressed genes are nevertheless still higher than in wtMYC cells. So relative to the amount of MYC protein, repression by MYCVD is clearly compromised- exactly in line with all biochemical data that show reduced, but not completely absent interaction of MYC VD to MIZ1. The fact that p19ARF levels go down with time in Supplementary Figure 6D is actually quite a concern for their experimental system (according to their model, ARF should be absent in GMYC1 cells from the get-go and definitely it should not go down when total levels of MYC go down, be it VD or wild-type). It is maybe also worth noticing that the other genes they see as being differentially expressed (cilium-related genes) are also MIZ1 target genes, as shown by Roussel and colleagues.

OK, we agree that MYCVD might be present in higher levels than wt MYC which could explain some of the findings presented. ARF is definitely not absent in GMYC1 cells. It needs to be there but at a lower level than in MYCN-driven tumors. We do agree with the reviewer that it should not go down when total levels of MYC goes down but if MYCVD is present at higher levels than wt MYC it can explain why ARF is even more suppressed when we do this switch that likely takes a few days to reach its maximum.

Worth noting, in our MYCVD experiments we transduced all cells with MYCVD in an unbiased fashion without selection or sorting cells. Thus, there might be selection processes involved that will promote expansion of MYC high cells leading to a decrease of ARF levels over time in culture. Blots generated show the relative ARF levels that indeed might be at different levels in the different cell lines.

Anyway, we are happy to remove these data (which was now done) as they will not be central for the story. Merely, these experiments were added in a previous version of this manuscript because a reviewer asked for them.

So based on the data, I think the authors need to remove the claim that this is MIZ1-independent phenomenon and - unless I misunderstand it - also Figure Supplementary 6D. Having done so, this will be a very interesting paper.

OK, we will agree on removing this statement and the figures generated (like Supp. Fig 6D) that would support such a claim. This is now done. The information regarding MIZ dependence in tumor development or in the potential suppression of ARF, albeit interesting, is not central to this story as it is presented here.